# Hypoxic conditioning in Parkinson's disease: randomized controlled multiple N-of-1 trials

Preclinical evidence suggests positive symptomatic and neuroprotective effects of hypoxic conditioning in Parkinson's disease (PD). This study (NCT05214287) investigated the safety, feasibility, short-term symptomatic and downstream effects of hypoxic conditioning in individuals with PD. 20 individuals with PD (mean age 62, 10 women, Hoehn-Yahr 1.5-3) completed randomized controlled double-blinded multiple N-of-1 trials. Each participant underwent five different 45-minute hypoxia interventions in duplicate: continuous hypoxia at $FiO_2$ 0.163 and 0.127, intermittent (five-minute intervals interspersed with normoxia) at $FiO_2$ 0.163 and 0.127, and placebo. Primary outcomes were safety and feasibility as measured by adverse events, vital parameter disturbances, participant-rated discomfort and feasibility questionnaires. Secondary outcomes were short-term participant-rated and assessor-rated symptom scores. Exploratory indicators of target engagement were serum erythropoietin, brain-derived neurotrophic factor (BDNF), glial fibrillary acidic protein (GFAP), neurofilament light-chain (NfL), platelet-derived growth factor-receptor-β (PDGFRβ) and cortisol. Secondary outcomes were evaluated using frequentist and Bayesian analysis. 20 participants completed the protocol. The trial met its primary endpoints for safety and feasibility. 95 adverse events occurred, including one moderate and three serious events. Adverse events were not dose-dependent and occurred at comparable incidence following hypoxia and placebo. Hypoxic conditioning was well-tolerated. Low-$F_IO_2$ protocols caused significant oxygen desaturations in two participants. Participants considered longer-term application feasible. Intermittent hypoxia at $F_IO_2$ 0.163 modestly improved most participant-rated symptoms for several hours compared to placebo, but not assessor-rated scales. One hour after intervention, serum markers did not differ between interventions. Hypoxic conditioning is safe and feasible in individuals with PD, and specific protocols may be associated with short-term symptom improvement. These findings inform and support follow-up studies of longer-term safety and efficacy of hypoxic conditioning.

Parkinson's disease (PD) is a chronic and progressive neurodegenerative disorder for which only symptomatic treatments are available. Despite significant advances in our understanding of the molecular and cellular mechanisms underlying PD, disease-modifying therapies that can halt or slow down the disease progression are still lacking. Therefore, there is an urgent need for novel therapeutic strategies.

Preclinical studies have suggested that moderate hypoxia provokes the release of survival-enhancing neurotransmitters, such as

✉ e-mail: bas.bloem@radboudumc.nl

**Fig. 1 | CONSORT diagram of study participants through the protocol.** OFF-medication without dopaminergic medication, PFT pulmonary function testing, ECG electrocardiogram, SaO2 arterial oxygen saturation.

dopamine from the substantia nigra[1–6]. Through induction of hypoxia inducible factor 1 (HIF-1), hypoxia activates tyrosine hydroxylase (TH), the main rate-limiting enzyme in the production of dopamine[7,8], leading to a rise in dopaminergic activity[5–10]. These short-term effects could offer symptomatic benefits, much like dopaminergic pharmacotherapy. In addition, converging evidence suggests that repeated exposure to moderate hypoxia induces an evolutionary conserved adaptive survival mechanism, termed hypoxic conditioning. Adaptive responses involve improved cellular energy metabolism, which in PD is disturbed by mitochondrial dysfunction, a subsequent reduction in oxidative stress and induction of adaptive plasticity. These observations suggest that, in addition to acute symptomatic effects, hypoxic conditioning might exert long-term neuroprotective effects[11–14], which we have recently reviewed[15]. The broad, pleiotropic working mechanism of hypoxia-mediated metabolic adaptations might have advantages over pharmacotherapeutic approaches, which more typically deploys a single-pathway paradigm to achieve disease modification in PD[16,17]. Targeting the hypoxia response pathway might be a promising novel treatment strategy in PD with the potential to alleviate symptoms, and which might ultimately modify the course of PD[11]. Hypoxic conditioning has been used in a variety of populations, including individuals with spinal cord injury and multimorbidity, without notable adverse effects[18–27]. Contrarily, chronic (intermittent) hypoxia, for example, during obstructive sleep apnea, promotes α-synuclein aggregation and mitochondrial dysfunction, and is associated with increased neurodegeneration[28,29]. This delicate balance between neuroprotection and neurodegeneration[29] raises questions about the safety and feasibility, and the optimal protocol for such strategies in PD. However, no studies have systematically investigated the safety, dose-response relation or short-term effects of hypoxic conditioning in PD in a randomized trial[30].

In this phase 1 trial, we assessed the safety, feasibility, and short-term effects of different individual-session protocols of hypoxic conditioning in individuals with PD. This trial employed a double-blinded, randomized placebo-controlled multiple N-of-1 design to assess different hypoxic conditioning protocols in all participants using Bayesian and frequentist analyses. This design is especially useful as it supports the dose-finding character of this study, and allows for a randomized intervention order in every individual participant so that participants act as their own control, thus allowing for the comparison

of (sub)acute symptom responses across interventions and within individuals.

## Results

Of 88 recruited individuals, 31 were eligible for on-site screening, and 20 participants started the study protocol. Non-eligibility was usually due to self-reported cardiac or pulmonary comorbidity and occasionally by the inability of OFF-medication assessment. Ten participants were excluded during the screening procedure due to an obstructive ($n = 3$) or restrictive ($n = 2$) pulmonary function test, arterial oxygen saturation (SaO2) < 80% at fraction of inspired oxygen (F$_1$O2) of 0.133 ($n = 3$), or heart rhythm abnormalities on electrocardiogram ($n = 2$). One participant dropped out after two interventions due to recurrence of paroxysmal atrial fibrillation (unlikely related to study procedures) and was replaced. This participant was not included in the secondary outcomes analysis (Fig. 1). Therefore, 20 individuals successfully completed all 200 interventions. In evaluating the success of blinding, 19.5% of interventions was guessed correctly by participants, which is equal to chance. Baseline demographics of all 20 individuals who completed the study are outlined in Table 1.

### Primary outcomes

In total, 95 AEs were reported by 21 participants (of whom one dropped out), of which 91 were mild (Fig. 2). Transient ischemic attack (TIA, severe AE), recurrence of atrial fibrillation (severe AE), severe hypertension (>180 mmHg systolic, severe AE) and fall from stairs (moderate AE) were the four moderate or severe adverse events. These four were assessed as unlikely to be related to the study intervention due to their timing and context. Both acute-onset severe AEs, i.e., TIA and atrial fibrillation, occurred after a placebo intervention. Atrial fibrillation occurred in a patient with (in retrospect) a positive history for palpitations, and occurred 1.5 weeks after the first hypoxia intervention. The TIA occurred 3 weeks after the screening procedure and one week after a placebo intervention. Nearly all adverse events were reported 1 h up to 3 days after the intervention, apart from discomfort or dyskinesia due to prolonged immobilization OFF-medication during the intervention. There was no relation between the number of AEs and any intervention protocol (range 16–20 AEs). AEs were most common in participants with Hoehn & Yahr (H&Y) 3 (average 5.7 AEs per participant, compared to 4.6 and 2.8 in H&Y 1.5–2 and H&Y 2.5, respectively). However, AE incidence rates were not significantly different between disease severity subgroups. Most common mild AEs were dyskinesia ($n = 20$, 21.1%), fatigue ($n = 15$, 15.8%), tremors ($n = 8$, 8.4%) and headache ($n = 8$, 8.4%). None of these AEs were exclusive to hypoxia interventions.

During the study protocol, 2 out of 200 interventions were interrupted due to individuals meeting the stop criterion for oxygen saturation (i.e., <80%). Equally, in nearly all cases, the lowest saturation during the screening intervention at FiO2 0.127 ON-medication (with medication) was equal to or higher than the lowest saturation during the study interventions in OFF (without medication). No hypoxia-induced disturbances meeting the stop criteria were observed for breathing frequency, heart rate or systolic and diastolic blood pressure, although heart rate demonstrated a dose-dependent increase (absolute mean difference 5 beats/min for FiO2 0.127 and 3 beats/min for FiO2 0.163 versus placebo, $P < 0.001$) directly after intervention initiation. These data are summarized in the Supplementary Materials. After stratification according to H&Y stage, there were no significant between-group differences in responses in systolic or diastolic blood pressure, heart rate or heart rate variability to mild or moderate levels of hypoxia (data not shown).

Median participant-reported scores (10-point scale) for dizziness (1.0, IQR 1–3), stress (1.1, IQR 1–2) and discomfort (1.4, IQR 1–3) were

## Table 1 | Demographic characteristics

| Characteristics | |
|---|---|
| **Total participants (*n*)** | 20 |
| **Women (*n*)** | 10 (50%) |
| **Age, yrs (mean ± SD)** | 62 ± 5.9 |
| **Disease duration, yrs (mean ± SD)** | 4.5 ± 2.5 |
| **H&Y stage (*n*, %)** | **MDS-UPDRS-III (median, IQR)** |
| 1.5–2 (8 (40%)) | 41 (28–54) |
| 2.5 (5 (25%)) | 45 (35–55) |
| 3 (7 (35%)) | 49 (34–64) |
| **Body mass index (mean ± SD)** | 24.7 ± 2.8 |
| **Ethnicity (*n*, %)** | |
| Caucasian | 19 (95%) |
| Persian | 1 (5%) |
| **Comorbidity (*n*, %)** | |
| Hypertension | 3 (15%) |
| Hypercholesterolemia | 2 (10%) |
| Myocardial infarction | 1 (5%) |
| Transient ischemic attack | 1 (5%) |
| **Medication** | |
| Levodopa | 13 (65%) |
| Dopamine agonists | 5 (25%) |
| MAO-B inhibitors | 0 |
| Antihypertensives | 5 (25%) |
| Statins | 3 (15%) |
| **Levodopa-equivalent daily dose (mean ± SD)** | 451.3 ± 448 |
| **PDQ-39** | 29.1 ± 21.3 |
| **Assessor-rated items (mean ± SD)** | |
| MDS-UPDRS part III | 44.7 (10.4) |
| MiniBES test | 23.5 (2.7) |
| Timed Up & Go Test | 6.2 (2.2) |

*H&Y* Hoehn and Yahr stage, *MDS-UPDRS-III* Movement Disorders Society—Unified Parkinson's disease rating scale Part III motor examination, *IQR* inter quartile range.

low across all protocols, with no difference between intervention protocols (Supplementary Materials).

Participants reported low levels of discomfort, shortness of breath, nausea and fatigue in the feasibility questionnaire (median 0.5/10, IQR 0–1). Participants considered the current duration of individual interventions (median 0.25/10, IQR 0–1) and higher-frequency interventions as feasible (median 0.5/10, IQR 0–2.25). Most participants would prefer at-home interventions (median 0.5/10, IQR 0–2.5), and disagreed with the need for supervision during at-home interventions (median 0.25/10, IQR 0–1.25). Results are summarized in the Supplementary Materials.

### Secondary outcomes

Two individuals had missing data for two interventions on participant-rated scales. Therefore, these two interventions in these individuals were excluded from frequentist analysis and individual effects Bayesian analysis. In the Bayesian analysis, however, these individuals contributed to the group analysis effect of all interventions. There was no difference in MDS-UDPRS measurements over time in random effects mixed models in participants or on group level ($\beta = 0.016$, $P = 0.88$), indicating no signs of serial correlation.

Singles sessions of intermittent hypoxia at $F_IO_2$ 0.163 improved participant-selected symptoms (0.57, 95% CI 0.23–0.92) and urge to take dopaminergic medication (0.48, 95% CI 0.11–0.86), but not global symptom impression (0.25, 95% CI −0.07 to 0.57, Fig. 3). However, this improvement did not meet our predefined minimal clinically important difference (MCID) of 0.75. Women reported significantly more symptomatic improvement on all three participant-rated symptoms compared to men (estimates between 0.39 and 0.90). Disease severity did not modify the observed effects. Adding brain-derived neurotrophic factor (BDNF) as an interaction term further strengthened the post-intervention improvement on participant-selected symptoms (0.68, 95% CI 0.32–1.04), but weakened the improvement on urge to take dopaminergic medication (0.40, 95% CI 0.00–0.80). BDNF response did not affect global symptom results (0.012, 95% CI −0.023 to 0.00).

In the Bayesian analysis, intermittent hypoxia at $F_IO_2$ 0.163 showed some symptomatic improvement above the pre-defined MCID of 0.75 compared to placebo on participant-rated scales. On the participant-selected symptom scale, mean group effects ranged between −0.35 (intermittent hypoxia at $F_IO_2$ 0.127) and 0.82

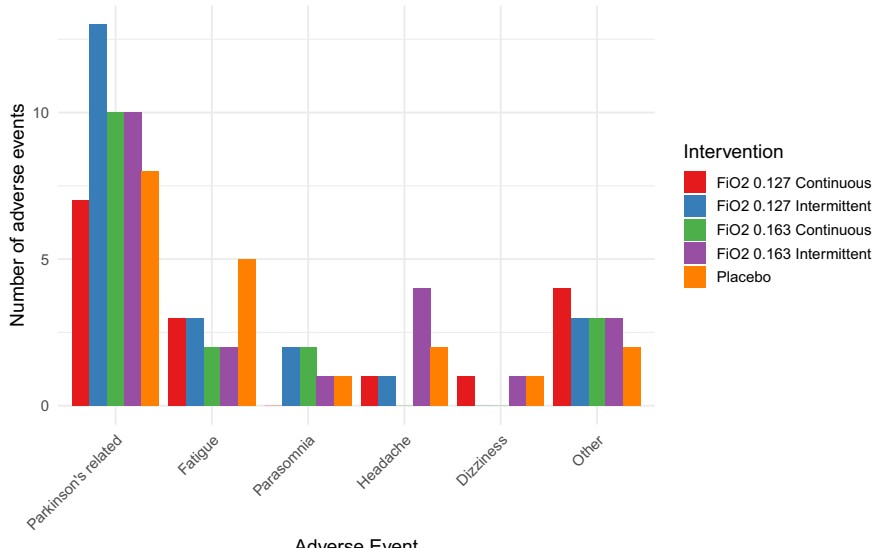

**Fig. 2 | Adverse events per intervention protocol and in categories.** "Other" adverse events included mostly self-limiting paresthesia in one or more extremity (6, occurring in both the placebo as active intervention group), vagal symptoms (2) and a sensitive throat (2).

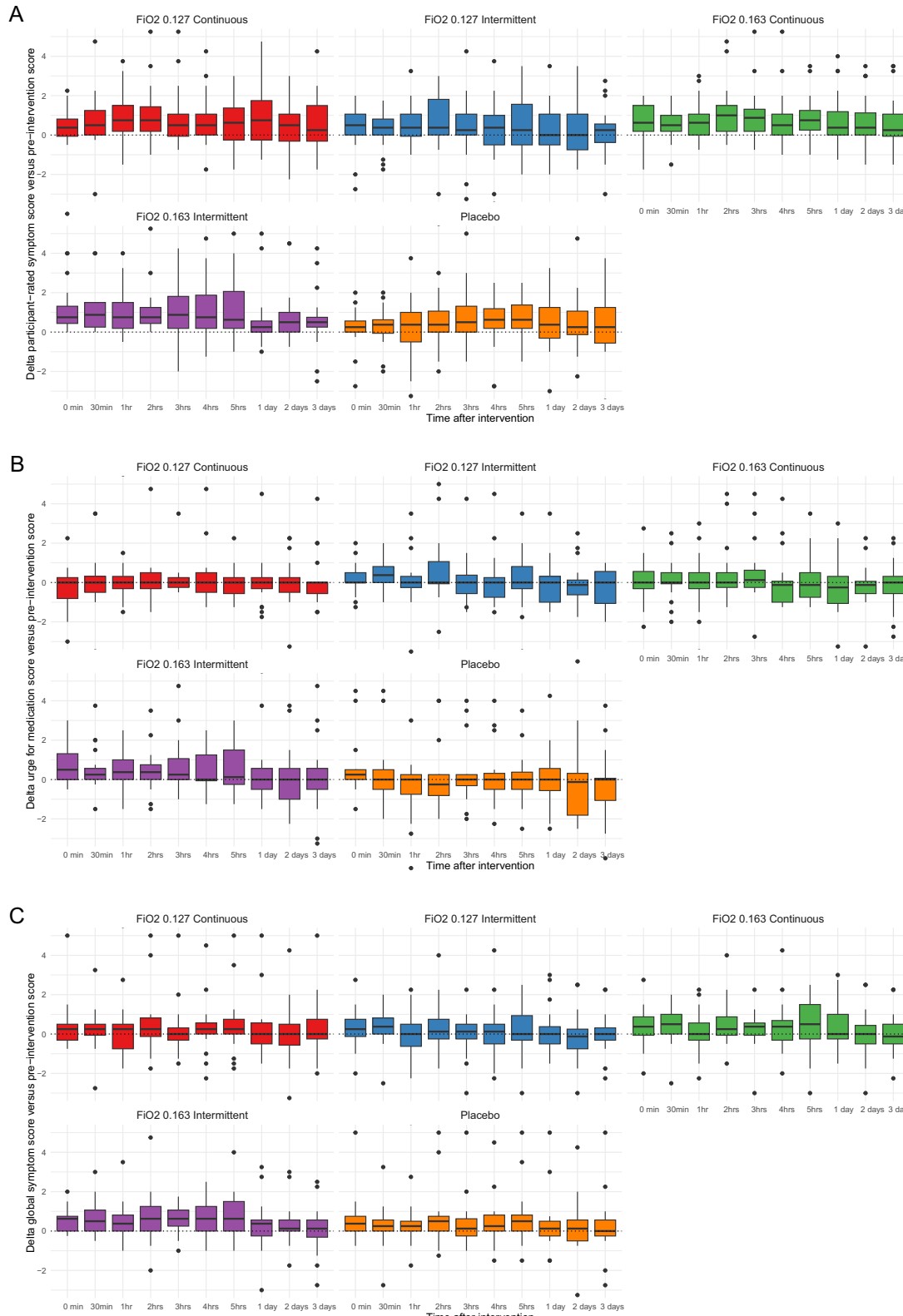

**Fig. 3 | Change in participant-reported symptoms after single sessions of hypoxic conditioning.** Change in the participant-selected symptom (usually the most prominent Parkinsonian symptom (**A**), urge to take dopaminergic medication (**B**) and global symptom impression (**C**). Three symptom scales were rated on a 10-point Likert scale pre- and post-intervention. The delta between the pre-intervention measurement and all subsequent post-intervention assessments is depicted in these figures. 0 min is the first post-intervention assessment, taken directly after the intervention, then 30 min post intervention, et cetera, until the last measurement 3 days post intervention.

(intermittent hypoxia at $F_IO_2$ 0.163) with individual probabilities from 12 to 98%. In terms of the urge to take dopaminergic medication, the differences ranged between 0.34 (continuous hypoxia at $FIO_2$ 0.163) and 0.81 (intermittent hypoxia at $aFIO_2$ 0.163), with individual probabilities from 0 to 94%. However, on the 10-point global symptom impression scale, mean group differences with placebo ranged between −0.81 (worst, intermittent hypoxia at $F_IO_2$ 0.127) and 0.01 (best, intermittent hypoxia at $F_IO_2$ 0.163). Individual probabilities for improvement above MCID ranged between 6 and 56%. PPT significantly improved compared to placebo after intermittent hypoxia at $F_IO_2$ 0.163 (0.87) and continuous hypoxia at $F_IO_2$ 0.127 (0.87) with high probabilities (>80%). TUGT, MDS-UPDRS part III and MiniBEST did not demonstrate acute symptomatic improvement above their respective MCIDs. Detailed frequentist and Bayesian analysis results are accessible online (https://github.com/Federica-Giardina/Talisman)[31].

With regard to assessor-rated scales, hypoxia protocols did not show significant improvement in Movement Disorders Society Universal Parkinson's Disease Rating Scale (MDS-UPDRS) part III relative to placebo ($P > 0.05$). Although intermittent hypoxia at $F_IO_2$ 0.163 reached the predefined MCID of 3.5 on the MDS-UPDRS part III, this difference was not significant ($P = 0.36$). Notably, the control intervention also improved 1.9 points on this scale. Purdue Pegboard Test (PPT, linear mixed models, $P = 0.99$), TUGT (Timed Up and Go Test, $P = 0.32$) and MiniBEST (Mini Balance Evaluation Systems Test, $P = 0.98$) did not differ between interventions.

### Exploratory outcomes

Although accelerometry-measured resting tremor decreased after hypoxia interventions, this was not significantly different from placebo (median 24.0% amplitude decrease versus pre-intervention, $P = 0.11$). Soluble platelet-derived growth factor receptor beta (PDGFrβ), erythropoietin (EPO), neurofilament light-chain (NfL) and BDNF were not changed 1 h post intervention (linear mixed models, $P > 0.05$ for all analyses). Glial fibrillary acidic protein (GFAP) increased across all groups (mean 29.6 pg/mL, $P < 0.001$), but there were no differences between subgroups. Apart from a significant circadian rhythm-mediated decrease in cortisol during all interventions (0.37–0.24 μmol/L, $P < 0.01$), there were no between-intervention differences ($P > 0.31$, Supplementary Materials).

## Discussion

To better understand the clinical potential for short-term exposure to hypoxic conditioning in individuals with PD, we performed this phase 1 trial, where we assessed the safety, feasibility, and short-term effects of different protocols. First, we demonstrated that the various protocols of continuous and intermittent hypoxic conditioning are safe and feasible in individuals with PD. Secondly, we observed that hypoxic conditioning protocols that reflect simulated altitude levels of 2000 and 4000 m do not evoke discomfort or a notable stress response. Finally, the intermittent hypoxia protocol at $F_IO_2$ 0.163 seemed most promising compared to the other hypoxia protocols for short-term improvement in participant-rated symptom scales, although the effects were modest. Also, assessor-rated assessment did not show significant acute improvement. With regard to mechanistic markers, no significant dose-dependent responses were identified. These observations warrant future studies to explore the potential clinical effects upon longer-term exposure to hypoxic conditioning in individuals with PD.

Following our primary aim, we found no differences in the number and nature of adverse events or participant-rated discomfort, stress and dizziness between the various hypoxic conditioning interventions, nor in comparison with the placebo intervention. Although continuous exposure to $F_IO_2$ 0.127 led to oxygen saturations below 80% for some participants, which was a stop criterion, this was not accompanied by any reported discomfort or other abnormal vital signs, supporting the notion that this intervention imposes limited physiological stress.

Following an amendment approved by the ethical committee, where we changed the protocol to assign either an $F_IO_2$ of 0.127 or 0.133 to any participant (see *Screening procedures*), several participants demonstrated relatively low $SaO_2$ levels of 80–85%. Nonetheless, these levels were not associated with treatment-specific adverse events, which suggests that single sessions of hypoxic conditioning appear to be safe in PD after thorough participant screening. Furthermore, we could not establish a difference in autonomic responses between disease severity groups, apart from a decreased hypoxic ventilatory response in people with PD[32]. This absence of hypoxia-related AEs is in line with previous trials in geriatric individuals[23,26], individuals with cardiovascular or pulmonary diseases[21–23,26,33], and dementia[12,34]. Although high-altitude illness can occur at $F_IO_2$ 0.16 or higher, our interventions were too brief as high-altitude illnesses usually arises after exposure of at least 1 h[35]. However, our results cannot be extrapolated to prolonged or repeated exposure to hypoxic conditioning. This is relevant as previous work found that after a 5-week protocol of intermittent hypoxia at $F_IO_2$ 0.10 for 1 h three times a week, transient angina and dizziness, dyspnea and headaches were reported in ~1% of a population of coronary artery disease patients[36]. In PD, a previous study investigated physiological responses to an unspecified 2-week protocol of intermittent hypoxia and reported no AEs, and an increase in serum dopamine[37,38]. Although participants in our study considered more frequent multiple-week hypoxic conditioning protocols feasible, follow-up studies are required to better understand the practical feasibility and longer-term clinical effects. Due to the individual differences in physiological responses to our hypoxia protocols, personalized dosing or gradually increasing levels of hypoxia may be considered.

Our secondary aim related to the effects of hypoxic conditioning on clinical outcomes, and we found that participant-rated symptom scales suggest modest short-term symptomatic effects of intermittent hypoxia at $F_IO_2$ 0.163 relative to placebo and other conditioning protocols, although the effects were not consistent across outcome measures. Although this dosage falls in the range of previous dose-finding studies that demonstrated a favorable benefit-harm ratios[39,40], most hypoxic conditioning studies have deployed $F_IO_2$ in lower ranges[41]. In animal studies, a short-lived but strong dose-response relation between $F_IO_2$ and striatal dopamine release was observed, suggesting a potential mediating mechanism for short-term effects[4]. In our study, we could not demonstrate such a dose-response relation in either symptoms or serum markers for target engagement. The observed duration of symptom improvement was in line with the first window of hypoxic conditioning, which suggests that effects subside after several hours[42]. It should be noted that we only investigated single hypoxic conditioning sessions and two different $F_IO_2$ levels. At least, our data suggests that the lowest $F_IO_2$ (0.127) could have been too intense, as evident from the tendency towards an increase in serum cortisol and the unexpectedly low oxygen saturation[43]. Although participant-rated symptom experience was not negative, maladaptive responses due to respiratory dysfunction might offset potential clinical and physiological benefits when applying $F_IO_2$ levels as low as 0.127. Another important observation is that the intermittent protocol (at $F_IO_2$ 0.163) showed significantly better effects than the continuous protocol at the same $F_IO_2$. The superiority of intermittent hypoxic conditioning over continuous hypoxic exposure is in line with previous observations in other (patient) populations[39,44]. Furthermore, preclinical studies indicate more potent HIF-1 upregulation after intermittent hypoxia[7,45]. Interestingly, women in our cohort reported more positive results on all participant-rated outcomes compared to men. Women are generally underrepresented in PD trials and a better understanding of gender differences is imperative. Factors to possibly consider include the notion that striatal binding on DAT scans is generally higher in women compared to men throughout the disease course, and also that levodopa-induced dyskinesias are more common in women (acknowledging that this could result directly from relative

overdosing in women)[46,47]. Although speculative, it is possible that a gender-related difference in striatal sensitivity to the short-term effects of hypoxic conditioning might explain some of the observed effect differences across genders. Future hypoxic conditioning studies should further investigate these effect differences.

Finally, serum PDGFRβ, EPO, BDNF, NfL, GFAP and cortisol were included as exploratory measures of target engagement and potential markers of neuroprotection and neuronal damage[48–52]. We found no significant increase in EPO and PDGFRβ 1 h post intervention, and variability in responses between individuals was high. This is the first study to investigate whether hypoxia induces relevant downstream pathways by measuring serum EPO and PDGFRβ in patients with PD. The lack of EPO increase in the short time frame measured of the current study is in line with previous findings in healthy adults[53]. Studies demonstrate that after a single session, serum EPO is only elevated after eight cycles of 4-min intermittent hypoxia[54], and that most prominent EPO release occurs hours up to 2 days post intervention[43,54–57]. In vitro, PDGFRβ reached peak expression after 6 h of hypoxia, with no significant increase after 1-h hypoxia[58,59]. Cortisol only showed a marginal signal towards an increase at the lowest $F_IO_2$ level, in accordance with earlier studies[60–62]. Therefore, short-term moderate hypoxia does not seem to affect hypothalamus-pituitary-adrenal axis activity in PD. This makes it unlikely that short-term symptomatic effects are mediated through stress systems such as the noradrenergic system, which is also implicated in PD symptom severity[63]. From a mechanistic point of view, some studies suggest that exercise and hypoxic conditioning have pathways in common, which we could not confirm for BDNF[14,64,65]. Several studies demonstrate that longer-term deep (intermittent) hypoxia is detrimental for mitochondrial dysfunction[66] and even induces α-synuclein aggregation and neurodegeneration[29]. Respiratory dysfunction in PD might accelerate neuronal hypoxic injury especially in this subgroup[32]. GFAP and NfL, as markers of neuronal stress and neural degeneration and potential biomarkers of PD progression[67,68], were not altered after single sessions of hypoxic conditioning, reflecting no acute neuronal injury within 2 h after onset of hypoxia. It should be noted that the first studies with these novel biomarkers suggest that in traumatic brain injury, an increase in serum becomes apparent only several hours post injury[69,70]. Future studies should examine whether these markers are affected after longer hypoxic conditioning protocols. On the other hand, activation of the hypoxia response pathway is linked to several PD risk genes, including LRRK2, DJ-1 and PINK1-Parkin[51]. Therefore, activation of using specific hypoxic conditioning protocols might have protective or compensatory properties. Future studies could investigate the effects of hypoxic conditioning on mitochondrial function and oxidative stress, such as by studying peripheral blood mononuclear cells (PBMCs).

This study has several strengths. The N-of-1 study design increases the power of the study, since participants serve as their own control. This has enabled both intra-individual as well as inter-individual effects analysis. Moreover, every treatment in the study was administered twice to reduce within-participant variance and increase power. An important limitation of the study includes the short time frame of the effect measurement. For example, first window effects may occur up to 24 h after hypoxic exposure[42], which are not covered by the present assessor-rated scales (such as MDS-UPDRS part III) and serum analyses, although the measurement of participant-reported symptom scores was extended beyond this window. Furthermore, the study had a small sample size of 20 participants. Whilst this reduces external validity, our observations highlight the need for follow-up studies and we strongly recommend such studies to include and acknowledge the heterogeneous nature of the PD population. We conducted a post-hoc sample size calculation for a future randomized control trial with intermittent hypoxia at $F_IO_2$ 0.163 and a control/placebo group (calculation publicly accessible at https://github.com/Federica-Giardina/Talisman). We first estimated the effect and precision using a mixed model adjusting for confounders (details below, Statistical Analysis section). For the precision, each individual contributed the average of two repeated measurements. A two-sample $t$-test with a significance level of 0.05 and 80% was used. This reveals that with observed effects on participant-reported outcomes, a sample size of $n = 45$ (for the participant-selected prime symptom) or $n = 158$ (for urge to take dopaminergic medication) per treatment group would be required in a future randomized-controlled trial. Lastly, our study focused on the immediate effects of a single exposure to hypoxic conditioning. The established effects on participant-rated outcomes were modest, yet clinically relevant, and warrant further investigation because of the low cost and burden of this non-pharmacological intervention. Therefore, future clinical and translational studies should focus on the clinical effects with higher-frequency and multiple-week interventions and mechanisms of hypoxic conditioning in PD, as well as explore potential small-molecule approaches towards hypoxia response cascade activation in PD.

## Methods
This single-center study was approved by the Medical Research Ethics Committee East Netherlands, The Netherlands (reference number NL.77891.091.22) and adheres to all relevant ethical regulations. It has been registered at Clinicaltrials.gov (ClinicalTrials.gov Identifier: NCT05214287). The detailed study protocol has been described elsewhere[71].

### Study objective
The primary objective was to evaluate the safety and feasibility of hypoxic conditioning in individuals with PD under controlled circumstances. Secondary objectives included acute symptomatic responses to hypoxic conditioning on participant-reported and standardized motor scales, which may provide valuable insight into identifying the best possible protocol for future studies. Our tertiary objective was to explore target engagement and involved mechanisms by measuring serum biomarkers.

### Study design
Multiple randomized, double-blinded, and placebo-controlled N-of-1 trials were performed at Radboud University Medical Center, Nijmegen, the Netherlands. All participants underwent four distinct hypoxia and one placebo protocol in duplicate. This exposure to ten interventions per participant in total allowed us to compare treatment effects at the inter- and intra-individual level.

### Study population
Twenty participants with clinically diagnosed PD (Hoehn & Yahr stage 1.5–3) without current cardiorespiratory comorbidity and without deep brain stimulation were included from a national PD research recruitment database. Other inclusion and exclusion criteria are summarized in Supplementary Materials.

As this is the first in-patient controlled trial into hypoxic conditioning in PD, the sample size was informed by a previous multiple N-of-1 trial experience[72] and consensus for feasibility studies[73]. We conducted a post-hoc sample size calculation (see "Discussion"). Participants provided written informed consent upon inclusion. Individuals with Hoehn & Yahr stages >3 were excluded due to too high participant burden of weekly OFF medication testing. Dropped-out participants were replaced.

### Screening procedure
Patients were enrolled between February 2022 and December 2022. During on-site screening, participants underwent pulmonary function testing (PFT) using spirometry, a carbon monoxide diffusion capacity test. Resting 12-lead electrocardiography was performed to screen for cardiac abnormalities. If no cardiorespiratory abnormalities were noted, participants were exposed to gradually decreasing $F_IO_2$ under

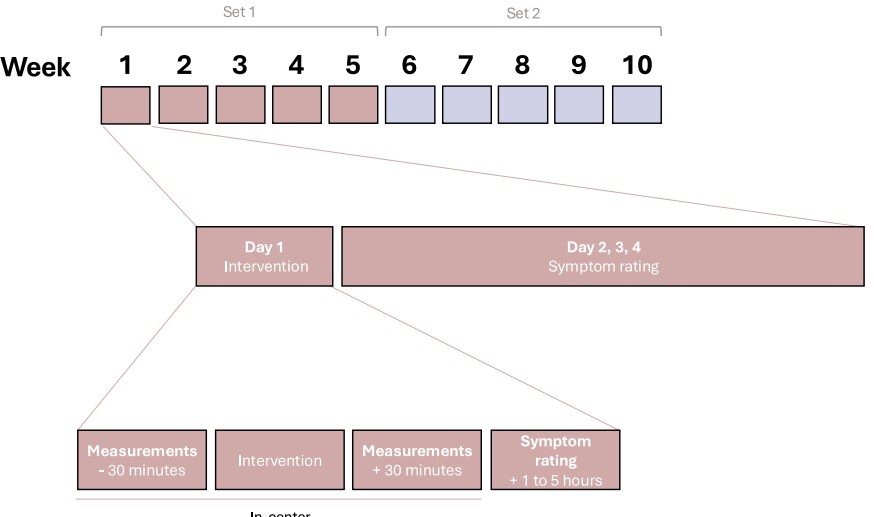

**Fig. 4 | Study protocol for every participant per week.** As an example, the contents of week 1 are highlighted. All subsequent intervention weeks are identical in events and measurements. The visits for Set 1 (in brown) and Set 2 (in blue) are identical in design and interventions, apart from each set being randomized separately.

arterial blood gas monitoring, until either an $F_IO_2$ of 0.127 or an $SaO_2 < 80\%$ was reached. At study initiation, individuals were excluded when $SaO_2 < 80\%$ was reached. During the screening procedure, a larger number of participants than expected based on the hypoxic conditioning literature demonstrated $SaO_2$ levels <80% before an $F_IO_2$ of 0.127 was reached. However, nearly all individuals demonstrated $SaO_2 \geq 80\%$ at a slightly higher $F_IO_2$ of 0.133. Therefore, the ethical committee approved a protocol amendment to additionally test participants at $F_IO_2$ 0.133 if $SaO_2$ dropped below 80% before an $F_IO_2$ 0.127 was reached. If $SaO_2$ remained above 80% at $F_IO_2$ 0.133, individuals were still eligible for inclusion, and the continuous and intermittent interventions at $F_IO_2$ 0.127 were administered at $F_IO_2$ 0.133. As a consequence, two individuals that were initially excluded, could now complete the protocol within stop criteria with this personalized dosage strategy, and six other individuals could now enter the protocol with interventions at $F_IO_2$ 0.133 instead of 0.127.

## Interventions

The spectrum of included hypoxia protocols was informed by multiple dose-finding hypoxic conditioning studies and included both mild and moderate hypoxic triggers[39,44,74–77]. Although the total hypoxic dosage in the intermittent hypoxia interventions is lower due to the in-between normoxic bouts, we were interested in comparing the same-duration interventions of continuous hypoxia in this dose-finding study and ensuring all other variables remain constant across interventions. Continuous hypoxia and placebo were included as comparators to intermittent hypoxia. Every participant received two sets of five different conditions in randomized order, consisting of four active interventions and one placebo, all with a 45-min total duration. Per week, one intervention was administered, adding up to a total intervention phase of 10 weeks:

- Continuous hypoxia at $FiO_2$ 0.163.
- Continuous hypoxia at $FiO_2$ 0.127 or 0.133.
- Intermittent hypoxia with $5 \times 5$ min at $FiO_2$ 0.163, interspersed with 5 min normoxic recovery.
- Intermittent hypoxia with $5 \times 5$ min at $FiO_2$ 0.127 or 0.133, interspersed with 5 min normoxic recovery.
- Continuous normoxia (placebo).

Interventions were administered using a commercially available hypoxic generator (*b-Cat ALT-120*, B-cat High Altitude, Tiel, the Netherlands). The machine lowers $FiO_2$ of room air by pressure swing adsorption. Hypoxic air is administered through a closed-circuit tubing system to a masked patient. Similar devices are regularly used in hypoxic conditioning studies[43,78–82].

Every week across 10 consecutive weeks, participants visited the hospital to receive one of the five possible interventions (Fig. 4). Period and carry-over effects were mitigated by the long wash-out period between interventions, as well as by implementing a new baseline measurement before the start of every intervention. This measurement was then used to calculate subsequent short-term symptomatic effects of that intervention.

## Randomization and blinding

Participants were equally divided in five groups with different intervention sequence according to a Latin square design with five "periods" of interventions. These periods were grouped in two separate sets, each consisting of a randomized sequence with five periods (Fig. 3). Randomization was conducted by a biostatistician (FG) using the *R* programming language.

The lab technician was not blinded to the interventions for safety and monitoring purposes and oversaw randomization and ascertained intervention exposure. Participant and outcome assessor were blinded. Success of blinding was evaluated by asking participants to guess the treatment assignment.

## Outcome measures

**Primary outcomes.** Safety was assessed for three domains: (1) the number, nature and severity of adverse events, (2) participants meeting stop criteria during the intervention, and (3) participant-rated dizziness, discomfort and stress on a 10-point Likert scale (allowing half points) every 10 min during interventions.

The blinded assessor and main investigator rated AEs in terms of severity based on the Common Terminology Criteria for Adverse Events (CTCAE) version 5.0[83] and relatedness to study interventions or procedures. AEs were assigned to the last intervention before AE onset. In case of serious AEs, the treating physician (cardiologist or internist) of the participant was consulted for an independent assessment of relatedness to study interventions or procedures. During the intervention, participants were continuously monitored for meeting any stop criteria (Supplementary materials).

Feasibility was assessed by a customized questionnaire after session one, five and ten (i.e., study completion). The questionnaire was informed by a widely used feasibility framework[84]. Items in each category (e.g., acceptability, expectancy) were subsequently inspired by a variety of healthcare feasibility questionnaires.

**Secondary outcomes.** Acute effects on PD symptoms were analyzed by participant-reported and assessor-rated standardized motor scales that were measured 30 min pre- and 30 min post intervention, unless indicated otherwise. This means that before every intervention, a novel baseline measurement of secondary and explorative outcomes was conducted. The post-intervention window overlaps with the first therapeutic window of conditioning effects[42].

Participant-reported motor outcomes included 10-point Likert scales (allowing half points) for an important participant-selected symptom, general symptom impression and the urge to take dopaminergic medication. The participant-selected symptom category was based on goal attainment scaling principles—the symptom had to be prominent to the participant, fluctuate to some extent, and changes in severity had to be conveniently apparent to the participant. Participants scored these scales pre-intervention, and after 30 min, hourly in the 5 h following intervention, and 1, 2 and 3 days after the intervention in the morning, when the second window of hypoxic conditioning wanes[42].

Assessor-rated scales in the OFF-state consisted of MDS-UPDRS-III, Purdue pegboard test, Timed Up & Go Test and the Mini Balance Evaluation Systems Test (MiniBEST). Additionally, accelerometry-measured resting tremor (*Move4*, movisens GmbH, Germany) was measured pre-intervention, during intervention, directly after intervention and 30 min after intervention.

**Exploratory outcomes.** Briefly, blood was drawn using venipuncture in a serum tube at five different timepoints: two times before intervention, separated by 30 min; one time directly after intervention; and then after 30 and 60 min post intervention. After refrigeration for 30–60 min, serum was centrifuged for 10 min at 2000 rcf, 4 °C. Supernatant serum was collected and stored at −80 °C until further analysis. For all analyses except cortisol, only the samples 30 min pre-intervention and 60 min post intervention were tested.

As exploratory serum markers of target engagement, dose-response effects and induced mechanisms, EPO and PDGFRβ were measured in serum pre-intervention and compared to 60 min post intervention. EPO is strongly activated in response to hypoxia in healthy controls[43,54] and less strongly in elderly[43,85,86], but the response in people with PD is unknown. PDGFRβ is a tyrosine-kinase receptor that is shed by brain pericytes in response to hypoxia[87]. Transient activation of its signaling pathway is associated with neuroprotective effects through PI3k/Akt activation[52,59,87]. As an additional exploratory marker of neuroprotection, analysis of BDNF was included. Preclinical and clinical studies suggest acute effects of hypoxia on BDNF[88,89] and BDNF levels are associated with symptom severity in PD, possibly reflecting reduced neuroplasticity[90,91]. Serum NfL and GFAP were included as acute markers of neuronal injury[92]. NfL and GFAP are both associated with symptom severity in PD[68,93–95]. Early evidence shows that exercise, a conditioning intervention with overlapping working mechanisms compared to hypoxic conditioning[15], reduces NfL[96]. Notably, both NfL and GFAP are induced by chronic intermittent hypoxia, such as occurring in sleep apnea[97,98]. These markers were analyzed using ultrasensitive single-molecule array (Simoa, Neurology 2-PLEX, Quanterix®, USA). Lastly, serum cortisol is included as a measure of the subacute stress response to hypoxia. Cortisol was measured 30 min and directly before intervention to take into account the circadian cortisol rhythm, and 0, 30, and 60 min post intervention.

Briefly, tremor as measured by wrist-worn accelerometry were analyzed using the publicly accessible Fieldtrip toolbox in MatLab. A time-frequency analysis was performed using (shifting) Hanning windows of five seconds with 2-s shifts. A conventional bandpass filter at 2–11 Hz was applied to specify PD resting tremor frequencies of interest. After taking the logarithm the power spectral density was calculated for all interventions for the peak frequency ±0.5 Hz to calculate the amplitude of tremor movement.

## Statistical analysis

Primary outcomes were analyzed using descriptive statistics. In accordance with our earlier aggregated N-of-1 experience[72], secondary outcomes were analyzed using linear mixed models as well as Bayesian analyses by aggregating 20 N-of-1 trials totaling 200 interventions. All secondary outcomes post-intervention were compared to the unique baseline measurements before every intervention to reduce period effects. We fitted a linear mixed-effects model to analyze participant-rated symptom outcomes across 20 patients, each observed over 7 time points and repeated over 2 sets. One individual had a missing measurement at 1 day follow-up for participant-rated scales, which was interpolated taking the average of the measurement directly before and the measurement directly after. The model includes five interventions, with intervention five (placebo) set as the reference. Fixed effects included intercept, time, interventions main effects and treatment-by-time interactions The model was adjusted for BDNF, gender and disease severity. $P$ values are 2-sided, and $P < 0.05$ was the significance threshold. This mixed model assumes a linear trend over time with individual-specific random slopes. While this is a sensible first approach for comparing overall treatment effects and their trajectories, it imposes a linear change over time. Given our uncertainty regarding when peak responses might occur, alternative specifications have been explored to capture potential non-linear trends. Due to high intra-individual variability, this approach did not improve the fit in a relevant way.

The Bayesian analysis allowed for an estimation of the posterior probability that any specific intervention protocol results in a clinically beneficial effect at both the group and individual level. In addition, these methods are useful in slowly progressive disease and in exploratory research phases[72,99]. The measure upon which secondary outcomes were compared across different intervention protocols in the Bayesian analysis was defined as the standardized area under the curve of the consecutive measurements (scores) from 30 min up until 1 day post intervention. These scores were calculated for the placebo and the four active intervention protocols. Specifically, we set a hierarchical model to account for both between-patient variability (through patient-level means) and within patient measurement error. This hierarchical structure borrows strength across patients to better estimate overall treatment effects. We assumed that outcomes followed a normal distribution with individual treatment means for the intervention protocols around overall intervention group means. Vague priors were assigned to the means, and uniform priors over a large range were defined for the standard deviations. We then computed the overall difference for each intervention protocol and the placebo, but also for each individual. Importantly, we derived the posterior probability of reaching the predefined MCID of 0.75 points between baseline and post-intervention. If this posterior probability was greater than 80%, interventions were considered effective. Interventions with posterior probabilities <20% were considered ineffective. The MCID for MDS-UPDRS part III was −3.25[100], and was four for the MiniBEST[101]. Bayesian analyses were performed using *JAGS* in R version 3.4.1. The code used for the frequentist and Bayesian analysis, including an explanation for the model, is publicly accessible (https://github.com/Federica-Giardina/Talisman).

### Reporting summary

Further information on research design is available in the Nature Portfolio Reporting Summary linked to this article.

## Data availability

The data generated in this study have been deposited in the Zenodo database of the Michael J Fox Foundation for Parkinson's Research (the study funder): https://zenodo.org/communities/mjff/ under 'TALISMAN-1: Phase 1 randomized controlled trial of hypoxic conditioning' and https://doi.org/10.5281/zenodo.16758910. The data are available under

restricted access by the Michael J. Fox Foundation, which will assess the relevance of the access request for the objectives of the Foundation and further research pursuits in the direction of the current study.

## Code availability

All code generated in this study as used for statistical analyses is provided through GitHub: https://github.com/Federica-Giardina/Talisman/.

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

## Acknowledgements

We thank Mr. Ivo Derks and Ms. Bregina Kersten for their unwavering support as lab technicians in this study. We thank Dr. Jeroen van Hees, Dr. Matthijs Kox and Dr. Bas Christiaan Stunnenberg for their methodological advice during study conceptualization, and thank Annelot Bakker, BSc., Iris Kersten and Rian Roelofs for conducting the additional serum analyses. This study is supported by the Michael J. Fox Foundation (MJFF, Therapeutic Pipelines Program, funding numbers MJFF-019201 and MJFF-025704). The Centre of Expertise for Parkinson & Movement Disorders was supported by a Centre of Excellence grant by the Parkinson Foundation.

## Author contributions

Conceptualization: J.M.J.D., M.J.M., S.M., D.H.J.T., and B.R.B. Methodology: J.M.J.D., M.J.M., F.G., P.N.A., D.H.J.T., and B.R.B. Investigation: J.M.J.D., M.J.M., D.H.J.T., F.G., and B.R.B. Visualization: J.M.J.D., M.J.M., S.M., D.H.J.T., and B.R.B. Funding acquisition: M.J.M., D.H.J.T., P.N.A., and B.R.B. Project administration: J.M.J.D., M.J.M., and B.R.B. Supervision: M.J.M., D.H.J.T., and B.R.B. Writing—original: J.M.J.D., M.J.M., and D.H.J.T. Writing—review & editing: J.M.J.D., M.J.M., F.G., S.M., P.N.A., D.H.J.T., and B.R.B.

## Competing interests

Bastiaan R. Bloem has received honoraria from serving on the scientific advisory board for Abbvie, Biogen, UCB, and Walk with Path; has received fees for speaking at conferences from AbbVie, Zambon, Roche, GE Healthcare, and Bial; and has received research support from The Netherlands Organisation for Scientific Research, the Michael J. Fox Foundation, UCB, Abbvie, the Stichting Parkinson Fonds, the Hersenstichting Nederland, the Parkinson Foundation, Verily Life Sciences, Horizon 2020, the Topsector Life Sciences and Health, and the Parkinson Vereniging. The other authors have nothing to disclose.

## Additional information

Jules M. Janssen Daalen ®[1,2], Marjan J. Meinders ®[1], Federica Giardina ®[3], Soania Mathur ®[4], Philip N. Ainslie[5],
Dick H. J. Thijssen[2,6] & Bastiaan R. Bloem ®[1] ✉

[1]Radboud University Medical Center; Donders Institute for Brain, Cognition and Behavior; Department of Neurology; Center of Expertise for Parkinson &
Movement Disorders, Nijmegen, The Netherlands. [2]Radboud University Medical Center, Department of Physiology, Nijmegen, The Netherlands. [3]Radboud
University Medical Center, IQ Health Department, Biostatistics Group, Nijmegen, The Netherlands. [4]UnshakeableMD, Oshawa, ON, Canada. [5]University of
British Columbia, Center for Heart, Lung and Vascular Health, School of Health and Exercise Sciences, Kelowna, BC, Canada. [6]Present address: Research
Institute for Sport and Exercise Sciences, Liverpool John Moores University, Liverpool, United Kingdom. ✉e-mail: bas.bloem@radboudumc.nl

