## [Transparent Peer Review file · Nature Communications]

Hypoxic conditioning in Parkinson's disease: randomized controlled multiple N-of-1 trials

Corresponding Author: Mr Jules Janssen Daalen

Version 1:

Reviewer comments:

Reviewer #1

(Remarks to the Author)

Note: This review was generated by two individuals that co-reviewed the manuscript and therefore combines their comments.

Introduction

This section could be improved. It would benefit from a discussion of the dual role of hypoxia as either causing neurodegenerative processes such as PD or allowing for neuroprotection. Currently, the authors only discuss the neuroprotection possibilities from hypoxia. The following recent references can be added to stress the point that hypoxia is a double-edged sword with PD.

- Hypoxia leads to increased aggregation of the protein alpha-synuclein, which has been implicated in Lewy body formation in PD – (PMID: 37860950, 35959288)

- Hypoxia leads to increased mitochondrial dysfunction, which is a hallmark of PD – (PMID: 37433770)

- Review article addresses this issue (PMID: 33862277)

By laying out this dichotomy in the introduction, a stronger argument is made for the author's Phase 1 study to determine the appropriate dose-response of hypoxic conditioning for PD.

Also, there should be more background on the nature of the aggregated N-of-1 design used and why it makes sense. This includes the ability to randomize the order of the interventions for any one patient given the short time it takes to administer them (the length of time it takes to administer the hypoxic challenge should be discussed as well) as well as the timing of hypoxia's effects; i.e., they must be acute, but how lingering are they? How long do the effects of any one hypoxic challenge last in terms of their further effects on clinical endpoints? In this context the authors should consider the possibility of carryover effects in conducting short term N-of-1 trials that have adjacent time periods in which different hypoxic challenges are administered.

Since there were no significant effects of hypoxia that would stand up to multiple comparisons adjustments, the authors should provide guidance in the discussion section on how large a study would be needed to adequately test the hypothesis of positive hypoxia effects based on the observed effect size in the N-of-1 trials. Also, the authors should address the clinical relevance of any assumed effect size.

Methods:

- Please give more detail regarding how you chose the sample size for this study. You state in the study population that the 'sample size was informed by a previous multiple N-of-1 trial experience' and cite the JAMA article, but this gives readers no real understanding of why you chose 20. Does this mean you just decided that since the JAMA article on mexiletine had 30 individuals, you would have 20?

- For the trial design and execution, there are quite a few things that are unclear. For example, how long was each intervention? 30-45 minutes? Were they all done on a single day or 1 on each of 5 different days over two weeks in which the 5 interventions were administered twice? Were the second week's interventions also randomized independently of the first week's? Since the interventions were given in a random order to each patient, the authors should simply make a figure with the basic layout in hours or days, and what went on for each patient in terms of the order of the interventions (e.g., 20 patients correspond to 20 rows in the table (with a header row with a title for each column), with each column reflecting the

order of the 10 interventions where each patient's cells in the table would contain what intervention they were provided in that time).

- I don't understand the Bayesian analysis and what it entails, so that should be described in much greater detail. Also, carryover effects as well as serial correlation and random effects should be explored (see comments below in the analysis subsection of the results section)

- Also, you state the study population was 'without cardiorespiratory comorbidity' but this is too vague. Did you exclude people with any diagnosis of coronary artery disease? Previous cerebrovascular accident or myocardial infarction? Arrhythmia? Confirmed diagnosis of asthma or COPD of any degree or just based on use of differing strengths of inhalers or inhaled steroids? Smokers vs non-smokers? Etc., etc. Please provide more details on these and any other important covariates or, if it these data weren't gathered, explain why they are missing from your study population.

- Did you consider measuring serum brain-derived neurotrophic factor (BDNF) as an exploratory outcome? BDNF levels are known to be decreased in PD and have been shown to increase in response to controlled hypoxia. (PMID: 17414812, 2229704, 24462831). If one MOA of hypoxia is to affect BDNF levels, then the proper analysis to link hypoxia to symptom improvement is to consider BDNF as a mediator in the analysis (see, e.g., PMID: 29283590)

Results:

- For Table 1, Demographic characteristics, please provide a more robust table with further details regarding characteristics that would be of particular importance to a study on hypoxic conditioning, as described in the comments above on Methods. Also include known use of any medications or deep brain stimulators for Parkinson's Disease.

• Key results: This paper reports the results of aggregated, randomized, double blinded, placebo controlled N-of-1 trials testing the safety, feasibility (primary outcomes) and symptom improvement (secondary outcome) of using different levels and lengths of hypoxic conditioning on patients with Parkinson's Disease. The authors show the short-term safety and feasibility of the different interventions. They also show that the hypoxic conditioning using the FiO₂ level of 0.163 is equal to placebo (normoxic) regarding assessor-reported symptom control, and superior to placebo in patient-reported symptom control. Finally, they showed that the N-of-1 trial design is efficient for performing randomized, controlled trials. The paper can really stimulate further interest, but not without providing more detail on various aspects of the design and analysis.

• Validity (evaluation of the validity and robustness of the data interpretation and conclusions): There are a few significant flaws in this manuscript that preclude publication, and they involve too little information provided about the study sample, the design of the study, and the analysis methods. The Results and Methods sections can be improved as described in detailed notes above.

• Originality and significance: This manuscript is original, both in its use of the aggregated N-of-1 trial design, and in its exploration of controlled hypoxia as a treatment for PD. As such, it holds significance for the broad scientific community because it is promoting novel clinical trial design, but also for neuroscientists and neurologists who study and treat PD. Unfortunately, many technical and data analysis details are missing or incomplete and need to be addressed.

• Data & methodology: Please comment on the validity of the approach, quality of the data and quality of presentation. Please note that we expect our reviewers to review all data, including any extended data and supplementary information. Is the reporting of data and methodology sufficiently detailed and transparent to enable reproducing the results? The methodology used is not detailed and needs attention. The authors should describe the Bayesian methods used and how they can deal with phenomena associated with N-of-1 trials: 1. Serial or autocorrelation between measurements over time; 2. Carry over effects; 3. Random effects. A mixed model could be used to test and address all these (PMID: 20863658; PMID: 35653405). Also, if some factors, such as BDNF levels are thought to mediate the relationship between hypoxia and clinical outcomes, then mediation analysis should be pursued (PMID: 29283590).

• Appropriate use of statistics and treatment of uncertainties: All error bars should be defined in the corresponding figure legends; please comment if that's not the case. Please include in your report a specific comment on the appropriateness of any statistical tests, and the accuracy of the description of any error bars and probability values. The authors should provide much more detail about analysis results, parameter estimates, and the statistical significance of the hypothesis tested. Also, they should consider using a correction for multiple comparisons, but acknowledge that in a phase I study links to efficacy are exploratory and can motivate follow-up studies that are appropriately powered based on the phase I results.

• How often was blood drawn? It is unclear in, e.g., Supplementary figure 12 exploring cortisol levels.

• Conclusions: Do you find that the conclusions and data interpretation are robust, valid and reliable? Not in current form. Hypoxia appears safe, but its broader effects are unclear based on a lack of detail of the analysis and analysis methods.

• Suggested improvements: Please list additional experiments or data that could help strengthening the work in a revision. Greater attention to the analysis and analysis methods is needed.

• References: Additional references to consider:

o Burtscher, PMID: 37860950 <https://physoc.onlinelibrary.wiley.com/doi/10.1113/JP285230>

o Burtscher, PMID: 33862277 <https://doi.org/10.1016/j.arr.2021.101343>

o Kalva-Filho, PMID: 37872033 DOI: 10.1016/j.parkreldis.2023.105885

o Ho, PMID: 37437730 DOI: 10.1038/s41467-023-39811-9

o Guo, PMID 35959288 DOI: 10.3389/fnagi.2022.919343

o Ferris, PMID: 17414812 DOI: 10.1249/mss.0b013e31802f04c7

o Vermehren-Schmaedick, PMID: 22297041 DOI: 10.1016/j.neuroscience.2012.01.017

o Helan, PMID: 24462831 DOI: 10.1016/j.yjmcc.2014.01.006

• Clarity and context: The abstract is clear and appropriate. The introduction can be improved as described above.

• Inflammatory material: There is no inflammatory material.

(Remarks to the Author)

Review of NCOMMS-24-09109A

- **Key results:** This paper reports the results of aggregated, randomized, double blinded, placebo controlled N-of-1 trials testing the safety, feasibility (primary outcomes) and symptom improvement (secondary outcome) of using different levels and lengths of hypoxic conditioning on patients with Parkinson's Disease. The authors show the short-term safety and feasibility of the different interventions. They also show that the hypoxic conditioning using the FiO₂ level of 0.163 is equal to placebo (normoxic) regarding assessor-reported symptom control, and superior to placebo in patient-reported symptom control. Finally, they showed that the N-of-1 trial design is efficient for performing randomized, controlled trials. The paper can really stimulate further interest, but not without providing more detail on various aspects of the design and analysis.
- **Validity:** There are a few significant flaws in this manuscript that preclude publication, and they involve too little information provided about the study sample, the design of the study, and the analysis methods. The Results and Methods sections can be improved as described in detailed notes above.
- **Originality and significance:** This manuscript is original, both in its use of the aggregated N-of-1 trial design, and in its exploration of controlled hypoxia as a treatment for PD. As such, it holds significance for the broad scientific community because it is promoting novel clinical trial design, but also for neuroscientists and neurologists who study and treat PD. Unfortunately, many technical and data analysis details are missing or incomplete and need to be addressed.
- **Data & methodology:** The methodology used is not detailed and needs attention. The authors should describe the Bayesian methods used and how they can deal with phenomena associated with N-of-1 trials: 1. Serial or autocorrelation between measurements over time; 2. Carry over effects; 3. Random effects. A mixed model could be used to test and address all these (PMID: 20863658; PMID: 35653405). Also, if some factors, such as BDNF levels are thought to mediate the relationship between hypoxia and clinical outcomes, then mediation analysis should be pursued (PMID: 29283590).
- **Appropriate use of statistics and treatment of uncertainties:** All error bars should be defined in the corresponding figure legends; please comment if that's not the case. Please include in your report a specific comment on the appropriateness of any statistical tests, and the accuracy of the description of any error bars and probability values. The authors should provide much more detail about analysis results, parameter estimates, and the statistical significance of the hypothesis tested. Also, they should consider using a correction for multiple comparisons, but acknowledge that in a phase I study links to efficacy are exploratory and can motivate follow-up studies that are appropriately powered based on the phase I results.
- **Conclusions:** In current form, the conclusions and data interpretation are not robust, valid or reliable. Hypoxia appears safe, but its broader effects are unclear based on a lack of detail of the analysis and analysis methods.
- **Suggested improvements:** Greater attention to the analysis and analysis methods is needed, as discussed above and in the detailed sections below.
- **References:** Additional references to consider:
 - 1) Burtscher, PMID: 37860950 <https://physoc.onlinelibrary.wiley.com/doi/10.1113/JP285230>
 - 2) Burtscher, PMID: 33862277 <https://doi.org/10.1016/j.arr.2021.101343>
 - 3) Kalva-Filho, PMID: 37872033 DOI: 10.1016/j.parkreldis.2023.105885
 - 4) Ho, PMID: 37437730 DOI: 10.1038/s41467-023-39811-9
 - 5) Guo, PMID 35959288 DOI: 10.3389/fnagi.2022.919343
 - 6) Ferris, PMID: 17414812 DOI: 10.1249/mss.0b013e31802f04c7
 - 7) Vermehren-Schmaedick, PMID: 22297041 DOI: 10.1016/j.neuroscience.2012.01.017
 - 8) Helan, PMID: 24462831 DOI: 10.1016/j.yjmcc.2014.01.006
- **Clarity and context:** The abstract is clear and appropriate. The introduction can be improved as described above.
- **Inflammatory material:** There is no inflammatory material.

SPECIFIC COMMENTS

Introduction

- This section could be improved. It would benefit from a discussion of the dual role of hypoxia as either causing neurodegenerative processes such as PD or allowing for neuroprotection. Currently, the authors only discuss the neuroprotection possibilities from hypoxia. The following recent references can be added to stress the point that hypoxia is a double-edged sword with PD:
 - Hypoxia leads to increased aggregation of the protein alpha-synuclein, which has been implicated in Lewy body formation in PD – (PMID: 37860950, 35959288)
 - Hypoxia leads to increased mitochondrial dysfunction, which is a hallmark of PD – (PMID: 37433770)
 - This review article addresses this issue (PMID: 33862277)
- By laying out this dichotomy in the introduction, a stronger argument is made for the author's Phase 1 study to determine the appropriate dose-response of hypoxic conditioning for PD.
- Also, there should be more background on the nature of the aggregated N-of-1 design used and why it makes sense. This includes the ability to randomize the order of the interventions for any one patient given the short time it takes to administer them (the length of time it takes to administer the hypoxic challenge should be discussed as well) as well as the timing of hypoxia's effects; i.e., they must be acute, but how lingering are they? How long do the effects of any one hypoxic challenge

last in terms of their further effects on clinical endpoints? In this context the authors should consider the possibility of carryover effects in conducting short term N-of-1 trials that have adjacent time periods in which different hypoxic challenges are administered.

Methods:

- Please give more detail regarding how you chose the sample size for this study. You state in the study population that the 'sample size was informed by a previous multiple N-of-1 trial experience' and cite the JAMA article, but this gives readers no real understanding of why you chose 20. Does this mean you just decided that since the JAMA article on mexiletine had 30 individuals, you would have 20?
- For the trial design and execution, there are quite a few things that are unclear. For example, how long was each intervention? 30-45 minutes? Were they all done on a single day or 1 on each of 5 different days over two weeks in which the 5 interventions were administered twice? Were the second week's interventions also randomized independently of the first week's? Since the interventions were given in a random order to each patient, the authors should simply make a figure with the basic layout in hours or days, and what went on for each patient in terms of the order of the interventions (e.g., 20 patients correspond to 20 rows in the table (with a header row with a title for each column), with each column reflecting the order of the 10 interventions where each patient's cells in the table would contain what intervention they were provided in that time).
- I don't understand the Bayesian analysis and what it entails, so that should be described in much greater detail. Also, carryover effects as well as serial correlation and random effects should be explored (see comments below in the analysis subsection of the results section)
- Also, you state the study population was 'without cardiorespiratory comorbidity' but this is too vague. Did you exclude people with any diagnosis of coronary artery disease? Previous cerebrovascular accident or myocardial infarction? Arrhythmia? Confirmed diagnosis of asthma or COPD of any degree or just based on use of differing strengths of inhalers or inhaled steroids? Smokers vs non-smokers? Etc., etc. Please provide more details on these and any other important covariates or, if it these data weren't gathered, explain why they are missing from your study population.
- Did you consider measuring serum brain-derived neurotrophic factor (BDNF) as an exploratory outcome? BDNF levels are known to be decreased in PD and have been shown to increase in response to controlled hypoxia. (PMID: 17414812, 2229704, 24462831). If one MOA of hypoxia is to affect BDNF levels, then the proper analysis to link hypoxia to symptom improvement is to consider BDNF as a mediator in the analysis (see, e.g., PMID: 29283590)

Results:

- For Table 1, Demographic characteristics, please provide a more robust table with further details regarding characteristics that would be of particular importance to a study on hypoxic conditioning, as described in the comments above on Methods. Also include known use of any medications or deep brain stimulators for Parkinson's Disease.
- How often was blood drawn? It is unclear in, e.g., Supplementary figure 12 exploring cortisol levels.

Discussion:

- Since there were no significant effects of hypoxia that would stand up to multiple comparisons adjustments, the authors should provide guidance in the discussion section on how large a study would be needed to adequately test the hypothesis of positive hypoxia effects based on the observed effect size in the N-of-1 trials. Also, the authors should address the clinical relevance of any assumed effect size.

Reviewer #3

(Remarks to the Author)

The present manuscript by Daalen et al. reported some novel findings from a Phase 1 trial investigating the effects of hypoxic conditioning on symptoms and serum biomarkers in 20 patients with Parkinson's Disease (PD). Overall, to my knowledge, this is the first randomized, double-blinded, placebo-controlled N-of-1 trial on a promising non-pharmacological intervention (i.e. controlled systemic intermittent hypoxia) for symptom control and disease modifying in PD patients. The apparent strengths of this group of authors include their expertise in clinical management of PD and design of N-of-1 trials (e.g. Ref. 56, JAMA 2018). In brief, this study represents a major advance to bring a largely experimental approach – hypoxic conditioning into a clinical reality in alleviating such a severe neuro-motor disorder – PD, which remains incurable with the current pharmacotherapy. Nevertheless, I have the following few concerns or suggestions for the authors to consider for further improvement of the manuscript.

1) Above all, while the goals in Primary Outcomes have been well achieved, these results in safety and adverse events are somewhat predictable, based on numerous previous clinical studies in various healthy or patient subjects. In addition, in 3rd Outcomes measures, two selected biomarkers – EPO and PDGFRb were largely unaffected by the single exposure to intermittent hypoxia. Could the authors perform additional analyses that may result in more mechanistic insights that will stimulate further investigation. For example, could you further check if other proposed biomarkers of PD could be modified by various regimens of hypoxic conditioning? Please consider the following biomarkers of interest: hypoxanthine (a purine metabolite altered by PD, see PMID: 32903216); neurofilament light (NFL) and glial fibrillary acid protein (GFAP) see PMID: 36997567; and C-reactive protein (see PMID: 31693246). Any positive findings in these new biomarkers will greatly enhance this work in terms of extraordinary innovation and quality required by Nature Communication.

- 2) The style and quality of Tables and Figures are not very impressive. For example, Table 1 appears oversimplified. More morphometric and medical information should be added to Table 1, for instance, racial background, BMI, medical history, including medication taken, etc.
- 3) The current in-text Table 1 and Figure 1 are not impressive. Do you think inclusion of Supplemental Figure 7 and/or Figure 8 as in-text figure(s) would be more appealing?
- 4) I would suggest using "Violin plots" to present the data instead of the current bar graphs in Supplementary Figure 3, 4, 5, 6. Violin plots also show each individual data points and improve data transparency.
- 5) In Supplementary Figure 11, please choose different types of symbols for each of the treatment conditions. This will help readers visually to differentiate the treatment conditions.
- 6) Adding a new figure to provide an illustrative description of the study protocol, group assignment and timelines in detail would help the readers' comprehension.
- 7) Although both male and female PD patients 50/50 participated in this study, the authors did present nor discuss any gender-dependent differences in response to hypoxic conditioning.
- 8) Numerous errors can be seen in the References. Please carefully proofread and correct the references.
- 9) A full description of each of the abbreviated terms should be done at their first appearance in the text, not at the end of the paper (Methods).

Reviewer #4

(Remarks to the Author)

This manuscript reports a preliminary clinical trial testing the safety and feasibility of continuous and intermittent exposures to mild-moderate hypoxia, administered over 45 minutes, in Parkinson's Disease patients. The hypoxia exposures were well tolerated and were not associated with increased incidence or severity of adverse events, compared with normoxia placebo. A strength of this study is the repeated measures N-of-1 design, in which each subject completed two sets of all 5 exposures (normoxia, 45-min continuous mild or moderate hypoxia, and 45-min intermittent mild or moderate hypoxia). The hypoxia sessions did not produce statistically significant improvements in the subjects' Parkinson's Disease symptoms, which is not surprising given that clinical studies of intermittent hypoxia intervention for heart failure [Saeed et al., J Card Fail 2012;18:387-91; PMID 22555269], prediabetes [Serebrovska et al., High Alt Med Biol 2019;20:383-91; PMID 31589074] and cognitive impairment [Wang et al., Am J Alzheimers Dis Other Demen 2020;35:1533317519896725] showed multiple IH sessions/week for several weeks were required for appreciable benefits. Nevertheless, this study establishes safety of intermittent hypoxia therapy in PD patients.

Major Comments

Was a power analysis done a priori, to determine if 20 subjects would provide sufficient statistical power (i.e. 1- β)? Please explain your rationale for applying 45 minutes of continuous hypoxia, rather than 25 minutes, which would match the hypoxia "dose" (i.e., 5 x 5 minutes) of the intermittent hypoxia protocol. Please include body mass index (BMI) in Table 1. The BMI range of the participants may be of interest since a recent meta-analysis revealed an association of Parkinson's disease with low BMI [Li et al., J Neurophysiol 2024;131:311-20; PMID 38264801].

Minor Comments

Several acronyms appear in the manuscript and supplemental materials. To assist the reader, nonstandard abbreviations must be defined where they first appear. Several (SaO₂, FiO₂, MDS-UPDRS, MCID, EPO) are defined in the Methods, which follows the other manuscript sections, and "TUGT" is defined in the supplemental materials. Please move these definitions to the first uses of the abbreviations. ON, OFF, PPT and LMM are not defined; please do so.

Abstract: Delete the 9th sentence ("20 participants..."); the 3rd sentence ("20 individuals with PD...") already presents that information.

Stop criteria subsection of Results: Delete "intervention" after screening, since screening is not an intervention.

Screening Procedure paragraph (Methods): The first sentence indicates spirometry is done by measuring CO diffusion capacity, but it isn't. Did you mean to say "...using spirometry and by conducting a carbon monoxide diffusion capacity test."?

Supplemental Table 1: Should the legend read "Green indicates greater or equal to..."?

Supplemental Table 2: "during screening" not "during screening day"

Supplemental Figures 1-11: Please indicate if the vertical bars represent SD, or SEM.

Supplemental Figure 2: Were the hypoxia exposures continuous, or intermittent?

Supplemental Figures 4-7: The Figure numbers in the legends are incorrect.

Reviewer #5

(Remarks to the Author)

This is an interesting work by Professor Bloem and colleagues, studying the effect of hypoxic conditioning in patients with Parkinson's disease (PD), in a phase 1, randomized controlled multiple N-of-1 trials. The intervention is novel in PD, as well as the trial design. The primary endpoint of the trial is to assess safety, feasibility and short-term effects of different protocols of hypoxia in PD. Each participant completed 4 sessions of different hypoxia protocols (two different levels of FiO₂, either intermittent or continuous) and one placebo, once weekly, and this was repeated twice. The order of the sessions differed among participants.

Please find here my comments and questions to the authors:

Introduction

- The first paragraph and the section "In addition, converging evidence suggests that repeat exposure to moderate hypoxia induces an evolutionary conserved adaptive survival mechanism, termed hypoxic conditioning. Adaptive responses involve improved cellular energy metabolism, which in PD is disturbed by mitochondrial dysfunction, a subsequent reduction in oxidative stress and induction of adaptive plasticity" miss references.

Methodology

- The selection and profile of participants is a bit unclear. Although mentioned in the previously published open-access protocol I think it would be informative to state the inclusion and exclusion criteria.
- It would also be nice to state that/if it was a single center study.
- Regarding demographics, it was great to see a 1:1 female:male ratio.
- How was sleep apnea assessed before inclusion?
- Could the authors provide a flowchart of patient inclusion, screening failures as well as dropouts and the corresponding reasons?
- Is smoking, vascular comorbidities, prior ischemic events taken into consideration?
- Would presence of dysautonomia be of interest to be taken into consideration in the stratification of patients, despite early phase design and safety primary outcome, since it is relevant to the rationale behind the intervention?
- Similarly, have the authors considered environmental exposures and genetic risk factors that are strongly related to hypoxia response pathway in PD?
- Treatment with MAOB inhibitors has been suggested to modulate mitochondria homeostasis – is that taken into consideration in the selection of participants?
- Trial participants have H&Y ranging from 1.5. to 3, thus reflecting a wide range of motor symptom severity. This may be an important parameter in the context of this type of intervention, so I was wondering if the authors have looked into the adverse event number and type in each H&Y category separately.
- I understand that cognitive assessment could not be included at the day of each intervention, in OFF medication state, but it would be a very interesting parameter to know at baseline, as well as after the completion of the trial protocol (in ON medication state).
- In comparison to studies in Alzheimer, the duration of intervention is shorter – compared to repeated intermittent hypoxia sessions daily, 3-5 days, for one or several weeks. Is there a specific rationale behind it?
- In secondary outcomes the authors state: "This post-intervention window (i.e. 30 minutes) overlaps with the first therapeutic window of conditioning effects". In the corresponding reference (nr 69) this window is defined from minutes to 24 hours. One may suggest that it can have been too early to have measured treatment effect after only 30 minutes.
- The exploratory outcome measures could have been expanded with markers of HIF-1 induction downstream effects, oxidative stress and mitochondrial function for a clearer view of biological response to intervention.

Results

- Can the authors provide some more details on how the SAEs were assessed unlikely associated to intervention (except for the fall from stairs)? It is stated that the SAEs TIA and AF occurred after the placebo session, but was placebo the first session in the protocol, or had they performed any of the hypoxia sessions before that?
- Regarding efficacy, I suggest that the authors soften the statements regarding improvement, as placebo was similar or even slightly superior to hypoxia intervention in several outcome measures.
- Differences in self-reported symptoms do not seem to be consistent, or dose- or intervention type-related, neither are reflected to the rater assessments, which further precludes strong conclusions regarding efficacy.

Discussion

- Would the authors consider elaborating on whether hypoxia induced pathology in PD is cause or consequence (or both) and how would that, in relation to this trial's results, inform the design of future trials?
- The authors state: "Cortisol only showed a marginal signal towards increase at the lowest FiO₂ level, in accordance with earlier studies. Therefore, any short-term symptomatic effects are unlikely mediated through subacute stress responses". Is this sentence correct, since stress would be expected to affect PD symptoms negatively?

Overall, it is a novel and important study with somewhat complicated protocol, among other reasons due to the number of interventions that cannot be truly "washed out", since hypoxia preconditioning may have long lasting biological effects. The protocol compliance is very good, and the results support the conclusion regarding safety.

Version 2:

Reviewer comments:

Reviewer #1

(Remarks to the Author)

The authors have adequately addressed our concerns. There was a great deal of material and discussion in the reviewer responses that is helpful and the reshaped manuscript makes it much easier to pull things together.

Reviewer #2

(Remarks to the Author)

Reviewer #3

(Remarks to the Author)

In this revised manuscript the authors have carefully respond to all my previous concerns and suggestions and made the necessary changes in texts, figures, and tables. I have no further concern on this much improved and important work.

Reviewer #4

(Remarks to the Author)

This manuscript reports a randomized, double-blinded, placebo-controlled phase 1 clinical trial assessing hypoxia intervention in patients with Hoehn-Yahr scale 1.5-3 indicating early to mod-stage Parkinson's Disease (PD). The subjects tolerated the hypoxia exposures, although several experienced adverse events, most of minimal concern. The rationale, methods and results are presented coherently, and the Discussion places the findings in context without overstating the implications. This study lays the groundwork for expanded clinical testing of hypoxia for PD.

The N-of-1 design is an important strength of this study. Both 45 min continuous and 5 x 5 min intermittent hypoxia were applied, at what could be considered mild (FIO₂ 0.163) and moderate (FIO₂ 0.127 or 0.133) intensities, in each subject, affording robust Bayesian and frequentist statistical analysis and comprehensive evaluation of hypoxia's safety and feasibility for PD therapy.

Regarding secondary outcomes, the hypoxia interventions did not produce clinically important improvements in PD symptoms, although the subjects, particularly the women, reported some relief in their symptoms. Although achieving symptomatic improvements was not the primary objective, substantial treatment effects might not be expected from only one session per week. The outcomes of this study will inform the design of the planned studies described in the last paragraph of the Discussion.

The modest increases in heart rate during hypoxia is physiological evidence that the hypoxia exposures impose very little stress. A comment along those lines could be added to the Discussion.

The more intense hypoxia exposures (FIO₂ = 0.127) seemed to provoke adverse responses in some subjects. For their planned trials the authors might consider gradually intensifying the hypoxia exposures over the first several sessions from initial FIO₂ of 0.163 to the ultimate FIO₂ of 0.127. Doing so might provoke physiological adaptations improving the subject's hypoxia tolerance, analogous to gradually increasing altitude to condition the body for mountain climbing.

Did any of the 95 adverse events occur during the sessions, as opposed to post-session? Were AEs monitored only during the 10-week study, or was monitoring extended a certain time post-study (beyond week 10?). That information would be valuable to physicians considering hypoxia treatment for their patients.

Another matter to consider is the possibility of synergy between hypoxia and PD medications, since they act by different and possibly complementary mechanisms.

The Screening Procedure subsection of the Methods states that two subjects were retained in the study by raising the lower FIO₂ to 0.133, implying only those two subjects breathed the slightly higher FIO₂. However, supplemental Table 2 shows the lowest FIO₂ was 0.133 for eight subjects. Please report that information.

Minor comment: Please provide a legend for Figure 1.

Reviewer #5

(Remarks to the Author)

In this phase 1 randomized controlled trial, the authors employ a multiple N-of-1 trial design to evaluate the safety and feasibility of hypoxic conditioning in individuals with Parkinson's disease (PD). This methodological approach - both in terms of the individualized trial design and the associated statistical analyses - is relatively novel and complex within the PD field. However, this can be viewed as a strength of the study, as it introduces a potentially more sustainable and personalized framework for early-phase clinical trials in this population.

The authors have made substabtil revisions in response to reviewr feedback, resulting in a significantly improved manuscript. The authors' transparency in sharing portions of their statistical code and providing power analyses for future studies is also appreciated. The study's conclusions are well supported by the data, and the interpretation of the results is reasonable. The methodology is sound, and the additional information provided enhances the reproducibility of the work, as does the prior publication of the trial's rationale and protocol in an open-access format. Finally, the authors have also considered, investigated, and discussed sex differences, which adds to the comprehensiveness of the analyses.

I have only a couple of minor remaining comments

- LEDD scores can be added in Table 1 for completeness regarding symptomatic treatment.

- In line 98-101, the authors state: "One participant dropped out after two interventions due to recurrence of paroxysmal atrial fibrillation (unlikely related to study procedures) and was replaced. This participant was not included in the secondary outcomes analysis (Figure 2)." However, Figure 2 shows 20 participants finally included, and 20 is the number stated throughout the report, so this dropout was not replaced, if I understand the statement correctly.
- In Figure 3, state with an asterisk or footnote which AEs are included under the term "other", as well as their frequency and intervention category.

Response to reviewers

General response

We thank the editor and all reviewers for their extensive work and constructive suggestions that have significantly improved this paper. We have been able to address all suggestions, and have performed additional analyses, as requested. First, we have conducted extra serum analyses to expand the therapeutic response overview using the single-molecule array (Simoa) analysis technique, namely neurofilament light chain (NfL) and glial fibrillary acidic protein (GFAP). Furthermore, we have tested whether the interventional response was potentially mediated by brain-derived neurotrophic factor (BDNF), using ELISA. Lastly, we have modified the statistical analyses as recommended by the reviewers, and we have substantially revised the manuscript throughout. Of course, we remain available for any further comments that you or the reviewers might have.

Reviewer comments

Reviewer #1

Introduction

- This section could be improved. It would benefit from a discussion of the dual role of hypoxia as either causing neurodegenerative processes such as PD or allowing for neuroprotection. Currently, the authors only discuss the neuroprotection possibilities from hypoxia. The following recent references can be added to stress the point that hypoxia is a double-edged sword with PD:

- Hypoxia leads to increased aggregation of the protein alpha-synuclein, which has been implicated in Lewy body formation in PD – (PMID: 37860950, 35959288)
- Hypoxia leads to increased mitochondrial dysfunction, which is a hallmark of PD – (PMID: 37433770)
- This review article addresses this issue (PMID: 33862277)

- By laying out this dichotomy in the introduction, a stronger argument is made for the author's Phase 1 study to determine the appropriate dose-response of hypoxic conditioning for PD.

Response: we appreciate this important suggestion and have revised the Introduction, which now reflects this delicate balance between neuroprotection and neurodegeneration and cites the recommended relevant articles.

Manuscript changes (p. 3):

"The broad, pleiotropic working mechanism of hypoxia-mediated metabolic adaptations might have advantages over pharmacotherapeutic approaches which more typically deploys a single-pathway paradigm to achieve disease modification in PD.^{16,17} Targeting the hypoxia response pathway might be a promising novel treatment strategy in PD with the potential to alleviate symptoms, and which might ultimately modify the course of PD.¹¹ Hypoxic conditioning has been used in a variety of populations, including individuals with spinal cord injury and multimorbidity, without notable adverse effects¹⁸⁻²⁷. Contrarily, chronic (intermittent) hypoxia, for example during obstructive sleep apnea, promotes α -synuclein aggregation and mitochondrial dysfunction, and is associated with increased neurodegeneration.^{28,29} This delicate balance between neuroprotection and neurodegeneration²⁹ raises questions about the safety and feasibility and the optimal protocol for such strategies in PD. However, no studies have systematically investigated the safety, dose-response relation or short-term effects of hypoxic conditioning in PD in a randomized trial.^{30"}

- Also, there should be more background on the nature of the aggregated N-of-1 design used and why it makes sense. This includes the ability to randomize the order of the interventions for any one patient given the short time it takes to administer them (the length of time it takes to administer the hypoxic challenge should be discussed as well) as well as the timing of hypoxia's effects; i.e., they must be acute, but how lingering are they? How long do the effects of any one hypoxic challenge last in terms of their further effects on clinical endpoints? In this context the authors should consider the possibility of carryover effects in conducting short term N-of-1 trials that have adjacent time periods in which different hypoxic challenges are administered.

Response: in both the revised Introduction and Methods section, we now more explicitly explain the rationale for using this aggregated N-of-1 approach. Furthermore, we have provided references for the timing of interventions in this trial and have clarified the timeline of the protocol in *Figure 1*. This Figure details the order of events in every week during which an intervention is organized. To evaluate the lingering of (sub)acute effects, participant-reported symptoms were rated for three days after every intervention, despite the first window of conditioning effects being up to one day. Indeed, we recognize that after a day, effects on participant-rated symptoms subside. We have implemented all these changes in the methods, results and discussion section.

Manuscript changes (Introduction, p. 3):

"In this phase 1 trial, we assessed the safety, feasibility, and short-term effects of different individual-session protocols of hypoxic conditioning in individuals with PD. This trial employed a double-blinded, randomized placebo-controlled multiple N-of-1 design to assess different hypoxic conditioning protocols in all participants using Bayesian and frequentist analyses. This design is especially useful as it supports the dose-finding character of this study, allows for a randomized intervention order in every individual participant and participants act as their own control, thus allowing for the comparison of (sub)acute symptom responses across interventions and within individuals."

Methods (p. 12-13):

"The Bayesian analysis allowed for an estimation of the posterior probability that any specific intervention protocol results in a clinically beneficial effect at both group and individual level. In addition, these methods are useful in slowly progressive disease and in exploratory research phases.^{72,100} The measure upon which secondary outcomes were compared across different intervention protocols in the Bayesian analysis was defined as the standardized area under the curve of the consecutive measurements (scores) from 30 minutes up until 1 day post intervention. These scores were calculated for the placebo and the 4 active intervention protocols. Specifically, we set a hierarchical model to account for both between-patient variability (through patient-level means) and within patient measurement error. This hierarchical structure borrows strength across patients to better estimate overall treatment effects. We assumed that outcomes followed a normal distribution with individual treatment means for the intervention protocols around overall intervention group means. Vague priors were assigned to the means, and uniform priors over a large range were defined for the standard deviations. We then computed the overall difference for each intervention protocol and the placebo, but also for each individual. Importantly, we derived the posterior probability of reaching the predefined MCID of 0.75 points between baseline and post-intervention. If this posterior probability was greater than 80%, interventions were considered effective. Interventions with posterior probabilities <20% were considered ineffective."

Discussion (p. 7):

"Although this dosage falls in the range of previous dose-finding studies that demonstrated a favorable benefit-harm ratios^{39,40}, most hypoxic conditioning studies have deployed F_{iO_2} in lower ranges.⁴¹ In animal studies, a short-lived but strong dose-response relation between F_{iO_2} and striatal dopamine release was observed, suggesting a potential mediating mechanism for short-term effects.⁴ In our study, we could not demonstrate such a dose-response relation in either symptoms or serum markers for target engagement. The observed duration of symptom improvement was in line with the first window of hypoxic conditioning, which suggests that effects subside after several hours.⁴² It should be noted that we only investigated single hypoxic conditioning sessions and two different F_{iO_2} levels."

Figure 1 is now clarified (p. 22):

"Figure 1: Study protocol for every participant per week. As an example, the contents of week 1 are highlighted. All subsequent intervention weeks are identical in events and measurements."

Methods:

- Please give more detail regarding how you chose the sample size for this study. You state in the study population that the 'sample size was informed by a previous multiple N-of-1 trial experience' and cite the JAMA article, but this gives readers no real understanding of why you chose 20. Does this mean you just decided that since the JAMA article on mexiletine had 30 individuals, you would have 20?

Response: As the primary outcomes of this study were safety and feasibility, a formal power calculation for the secondary or exploratory outcomes were not previously part of the originally submitted manuscript. However, in answer to your question, we have performed a post-hoc sample size calculation, which was informed by our previous experience and assumptions based on the minimal clinically important difference for our participant-rated secondary outcomes. We have now added this clarification to the revised manuscript.

Discussion (p. 8):

We conducted a post-hoc sample size calculation for a future randomized control trial with intermittent hypoxia at F_{iO_2} 0.163 and a control/placebo group (calculation publicly accessible at <https://github.com/Federica-Giardina/Talisman>). We first estimated the effect and precision using a mixed model adjusting for confounders (details below, Statistical Analysis section). For the precision, each individual contributed the average of 2 repeated measurements. A two-sample t-test with a significance level of 0.05 and 80% was used. This reveals that with observed effects on participant-reported outcomes, a sample size of n=45 (for the participant-selected prime symptom) or n=158 (for urge to take dopaminergic medication) per treatment group would be required in a future randomized-controlled trial. Lastly, our study focused on the immediate effects of single exposure to hypoxic conditioning. Therefore, future clinical and translational studies should focus on the clinical effects with higher-frequency and multiple-week interventions and mechanisms of hypoxic conditioning in PD, as well as explore potential small-molecule approaches towards hypoxia response cascade activation in PD.

- For the trial design and execution, there are quite a few things that are unclear. For example, how long was each intervention? 30-45 minutes? Were they all done on a single day or 1 on each of 5 different days over two weeks in which the 5 interventions were administered twice? Were the second week's interventions also randomized independently of the first week's? Since the interventions were given in a random order to each patient, the authors should simply make a figure with the basic layout in hours or days, and what went on for each patient in terms of the order of the interventions (e.g., 20 patients correspond to 20 rows in the table (with a header row with a title for each column), with each column reflecting the order of the 10 interventions where each patient's cells in the table would contain what intervention they were provided in that time).

Response: we have clarified the study protocol in our revised manuscript. Although a detailed protocol paper on our study was published previously, we appreciate this reviewer's comments and have now clarified several elements of our methods. Briefly, participants visited the hospital *ten* times to receive an intervention. All measurements were conducted during every intervention day, as noted in the new *Figure 1*. Interventions took place across ten consecutive weeks, with a wash-out period between any two interventions of at least five days. This assures minimization of any possible effect lingering. Moreover, potential carry-over effects were minimized by randomizing the order of interventions between individuals. For most participants, this protocol with weekly interventions consisted of ten weeks (occasionally slightly longer, e.g. due to a vacation in-between sessions). A Latin square design was adapted for randomization, and randomization was indeed conducted in two separate blocks for the first and second half of the study protocol. All interventions lasted 45 minutes.

Manuscript changes:

“Every participant received two sets of five different conditions in randomized order, consisting of four active interventions and one placebo, all with a 45-minute total duration. Per week, one intervention was administered, adding up to a total intervention phase of 10 weeks:

- *Continuous hypoxia at FiO₂ 0.127 or 0.133.*
- *Intermittent hypoxia with 5x5 minutes at FiO₂ 0.127 or 0.133, interspersed with 5 minutes normoxic recovery.*
- *Continuous hypoxia at FiO₂ 0.163.*
- *Intermittent hypoxia with 5x5 minutes at FiO₂ 0.163, interspersed with 5 minutes normoxic recovery.*
- *Continuous normoxia (placebo)”*

“Every week across ten consecutive weeks, participants visited the hospital to receive one of the 5 possible interventions (Figure 1). Period and carry-over effects were mitigated by the long wash-out period between interventions, as well as by implementing a new baseline measurement before the start of every intervention. This measurement was then used to calculate subsequent short-term symptomatic effects of that intervention.”

“Participants were equally divided in five groups with different interventions sequence according to a Latin square design with five ‘periods’ of interventions. These periods were grouped in two separate sets, each consisting of a randomized sequence with five periods (Figure 1). Randomization was conducted by a biostatistician (FG) using the R programming language.”

- I don't understand the Bayesian analysis and what it entails, so that should be described in much

greater detail. Also, carryover effects as well as serial correlation and random effects should be explored (see comments below in the analysis subsection of the results section)

Response: we have now made the explanation of the Bayesian methods detailed and have made the code of the analysis available, enabling replication of our results. We have conducted the suggested additional exploratory analyses for serial correlation for the MDS-UPDRS part III (as we have not repeated these measurements beyond the expected effective time window, as we have with participant-rated symptoms), and demonstrate no such effects.

Manuscript changes:

Results (p. 6):

There was no difference in MDS-UDPRS measurements over time in random effects mixed models in participants or on group level ($\beta = 0.016$, $P = 0.88$), indicating no signs of serial correlation.

Methods, Interventions (p. 10):

Period and carry-over effects were mitigated by the long wash-out period between interventions as well as implementing a new baseline measurement before the start of every intervention, which is then used to calculate short-term symptomatic effects of that intervention.

Methods, Outcome measures (p. 11):

“Acute effects on PD symptoms were analyzed by participant-reported and assessor-rated standardized motor scales that were measured 30 minutes pre- and 30 minutes post-intervention, unless indicated otherwise. This means that before every intervention, a novel baseline measurement of secondary and explorative outcomes was conducted.”

Methods, Statistical analysis (p. 11-12):

In accordance with our earlier aggregated N-of-1 experience⁶⁶, secondary outcomes were analyzed using linear mixed models as well as Bayesian analyses by aggregating 20 N-of-1 trials totaling 200 interventions. All secondary outcomes post-intervention were compared to the unique baseline measurements before every intervention to reduce period effects. We fitted a linear mixed-effects model to analyze secondary outcomes across 20 patients, each observed over 7 time points and repeated over 2 sets. One individual had a missing measurement at one day follow-up for participant-rated scales, which was interpolated taking the average of the measurement directly before and the measurement directly after. The model includes 5 interventions, with intervention 5 (placebo) set as the reference. Fixed effects included intercept, time, interventions main effects and treatment-by-time interactions. The model adjusted for BDNF, gender and disease severity. P values are 2-sided, and $P < 0.05$ was the significance threshold. We included a random slope for time at the patient level, allowing the time effect to vary across individuals. This accounts for patient-specific variability in the change of outcomes over time.

The Bayesian analysis allowed for an estimation of the posterior probability that any specific intervention protocol results in a clinically beneficial effect at both group and individual level. In addition, these methods are useful in slowly progressive disease and in exploratory research phases.^{66,95} The measure upon which secondary outcomes were compared across different intervention protocols in the Bayesian analysis was defined as the standardized area under the curve of the consecutive measurements (scores) from 30 minutes up until 1 day post intervention. These scores were calculated for the placebo and the 4 active intervention protocols. Specifically, we set a hierarchical model to account for both between-patient variability (through patient-level means) and within patient measurement error. This hierarchical structure borrows strength across patients to better estimate overall treatment effects. We assumed that outcomes followed a Normal distribution with individual treatment means for the intervention protocols around overall intervention group means. Vague priors were assigned to the means, and uniform priors over a large range were defined for the standard deviations. We then computed the overall difference for each intervention protocol and the placebo, but also for each individual. Importantly, we derived the posterior probability of reaching the predefined MCID of 0.75 points between baseline and post-intervention. If this posterior probability was greater than 80%, interventions were considered effective. Interventions with posterior probabilities <20% were considered ineffective.

- Also, you state the study population was ‘without cardiorespiratory comorbidity’ but this is too vague. Did you exclude people with any diagnosis of coronary artery disease? Previous cerebrovascular accident or myocardial infarction? Arrhythmia? Confirmed diagnosis of asthma or COPD of any degree or just based on use of differing strengths of inhalers or inhaled steroids? Smokers vs non-smokers? Etc., etc. Please provide more details on these and any other important covariates or, if it these data weren’t gathered, explain why they are missing from your study population.

Response: we excluded all individuals with a confirmed current diagnosis of asthma or COPD (independent of current treatment), as well as individuals with abnormal pulmonary function testing (obstructive, restrictive or diffusion deficits). Furthermore, we excluded individuals with congestive heart failure and cardiac arrhythmia, including atrial fibrillation. Current smokers were excluded, and former smokers were included only if they had

normal pulmonary function tests. We have now clarified this in the revised Methods section and Supplementary Materials and have made available extensive demographic characteristics in *Table 1*.

Manuscript changes (p. 8):

“Twenty participants with clinically diagnosed PD (Hoehn & Yahr stage 1.5-3) without current cardiorespiratory comorbidity and deep brain stimulation, were included from a national PD research recruitment database. Other inclusion and exclusion criteria are summarized in Supplementary Materials.”

- Expanded Table 1 containing a detailed description of demographic characteristics.
- Added a new table: *Supplementary Table 1* (p. 23) with all inclusion and exclusion criteria.

- Did you consider measuring serum brain-derived neurotrophic factor (BDNF) as an exploratory outcome? BDNF levels are known to be decreased in PD and have been shown to increase in response to controlled hypoxia. (PMID: 17414812, 2229704, 24462831). If one MOA of hypoxia is to affect BDNF levels, then the proper analysis to link hypoxia to symptom improvement is to consider BDNF as a mediator in the analysis (see, e.g., PMID: 29283590)

Response: we have indeed considered this. We had previously decided to not include an analysis of serum BDNF, as we expected limited effects due to the short timeframe of the outcome measurement in this study. However, as some clinical and preclinical studies indeed suggest a possible short-term effect on BDNF, we have now included serum BDNF as an additional mechanistic outcome measure. This has now been added to the Results and Discussion. The complete analyses are reported in the Supplementary Materials.

Manuscript changes (Results, p. 5):

“Adding BDNF as an interaction term further strengthened the post-intervention improvement on self-selected symptoms (0.68, CI 95% 0.32-1.04), but weakened the improvement on urge to take dopaminergic medication (0.40, CI 95% -0.002–0.80). BDNF response did not affect global symptom results (0.012, 95% CI -0.023–0.00).”

Methods: “As an additional exploratory marker of neuroprotection, analysis of BDNF was included. Preclinical and clinical studies suggest acute effects of hypoxia on BDNF^{83,84} and BDNF levels are associated with symptom severity in PD, possibly reflecting reduced neuroplasticity.^{85,86}

[...]

As explorative analyses, BDNF, gender and disease severity were added as covariates.”

Results:

- For Table 1, Demographic characteristics, please provide a more robust table with further details regarding characteristics that would be of particular importance to a study on hypoxic conditioning, as described in the comments above on Methods. Also include known use of any medications or deep brain stimulators for Parkinson’s Disease.

Response: we have significantly expanded the table to now provide a more detailed characteristic of the study population, including detailed information regarding medication use. An active deep brain stimulation system was a contra-indication in this study, as we regarded the frequent (i.e., weekly) OFF-medication assessments too burdensome for this subgroup of patients.

Manuscript changes (Methods, p. 9):

“Twenty participants with clinically diagnosed PD (Hoehn & Yahr stage 1.5-3) without current cardiorespiratory comorbidity and without deep brain stimulation were included from a national PD research recruitment database. Other inclusion and exclusion criteria are summarized in Supplementary Materials.”

- Expanded Table 1 containing a detailed description of demographic characteristics.

• Key results: This paper reports the results of aggregated, randomized, double blinded, placebo controlled N-of-1 trials testing the safety, feasibility (primary outcomes) and symptom improvement (secondary outcome) of using different levels and lengths of hypoxic conditioning on patients with Parkinson’s Disease. The authors show the short-term safety and feasibility of the different interventions. They also show that the hypoxic conditioning using the FiO₂ level of 0.163 is equal to placebo (normoxic) regarding assessor-reported symptom control, and superior to placebo in patient-reported symptom control. Finally, they showed that the N-of-1 trial design is efficient for performing randomized, controlled trials. The paper can really stimulate further interest, but not without providing more detail on various aspects of the design and analysis.

Response: we thank both reviewers for their thoughtful comments and suggestions, to which we respond below.

• **Validity:** There are a few significant flaws in this manuscript that preclude publication, and they involve too little information provided about the study sample, the design of the study, and the analysis methods. The Results and Methods sections can be improved as described in detailed notes above.

Response: we have substantially revised the Methods section, paying special attention to the sample size calculation, the complex study design and corresponding statistical analyses, as suggested. A detailed description of our methods was published previously (PMID 35836147, <https://doi.org/10.1186/s12883-022-02770-7>), and we relied on this when we submitted the first version. We now appreciate that this likely resulted in a methods section in our original submission that contained too little detail on the methods. This also refers to some of the comments on the lack of detail listed below. We obviously understand that sufficient detail must be present in the present manuscript to improve readability and providing insight. Specific adjustments are discussed in detail below.

• **Originality and significance:** This manuscript is original, both in its use of the aggregated N-of-1 trial design, and in its exploration of controlled hypoxia as a treatment for PD. As such, it holds significance for the broad scientific community because it is promoting novel clinical trial design, but also for neuroscientists and neurologists who study and treat PD. Unfortunately, many technical and data analysis details are missing or incomplete and need to be addressed.

Response: we have addressed these concerns by providing more details on the analytical methods and have provided the code including instructions for use. Specific adjustments are discussed in detail below.

• **Data & methodology:** The methodology used is not detailed and needs attention. The authors should describe the Bayesian methods used and how they can deal with phenomena associated with N-of-1 trials: 1. Serial or autocorrelation between measurements over time; 2. Carry over effects; 3. Random effects. A mixed model could be used to test and address all these (PMID: 20863658; PMID: 35653405). Also, if some factors, such as BDNF levels are thought to mediate the relationship between hypoxia and clinical outcomes, then mediation analysis should be pursued (PMID: 29283590).

Response: led by our co-author FG, who is a statistician with special expertise in Bayesian analysis and related models, we now provide explanations for our Bayesian methods and how issues with N-of-1 trials were addressed. Furthermore, we have carried out the recommended mixed model analyses to mitigate these concerns. Specific adjustments are discussed in detail below.

• **Appropriate use of statistics and treatment of uncertainties:** All error bars should be defined in the corresponding figure legends; please comment if that's not the case. Please include in your report a specific comment on the appropriateness of any statistical tests, and the accuracy of the description of any error bars and probability values. The authors should provide much more detail about analysis results, parameter estimates, and the statistical significance of the hypothesis tested. Also, they should consider using a correction for multiple comparisons, but acknowledge that in a phase I study links to efficacy are exploratory and can motivate follow-up studies that are appropriately powered based on the phase I results.

Response: these points are all well taken. We have inspected all effect estimates and corresponding error bars and legends, and have complemented all descriptions where necessary. As suggested by these and other reviewers, we have now modified the frequentist section of our statistical analysis and have provided a rationale for this (see also additional comments below). We now also report the outcome of our results with correction for multiple comparisons, and have added an explicit interpretation of our results to the Discussion. Specific adjustments are discussed in detail below.

• **Conclusions:** In current form, the conclusions and data interpretation are not robust, valid or reliable. Hypoxia appears safe, but its broader effects are unclear based on a lack of detail of the analysis and analysis methods.

• **Suggested improvements:** Greater attention to the analysis and analysis methods is needed, as discussed above and in the detailed sections below.

Response: we have substantially revised the Methods section, paying special attention to sample size calculation, the complex study design and corresponding extra analyses. Specific adjustments are discussed in detail below.

• **References:** Additional references to consider:

- 1) Burtscher, PMID: 37860950 <https://physoc.onlinelibrary.wiley.com/doi/10.1113/JP285230>
- 2) Burtscher, PMID: 33862277 <https://doi.org/10.1016/j.arr.2021.101343>

- 3) Kalva-Filho, PMID: 37872033 DOI: 10.1016/j.parkreldis.2023.105885
- 4) Ho, PMID: 37437730 DOI: 10.1038/s41467-023-39811-9
- 5) Guo, PMID 35959288 DOI: 10.3389/fnagi.2022.919343
- 6) Ferris, PMID: 17414812 DOI: 10.1249/mss.0b013e31802f04c7
- 7) Vermehren-Schmaedick, PMID: 22297041 DOI: 10.1016/j.neuroscience.2012.01.017
- 8) Helan, PMID: 24462831 DOI: 10.1016/j.yjmcc.2014.01.006

Response: thank you for highlighting this important body of work. We have now added all suggested references to the revised manuscript.

• **Clarity and context: The abstract is clear and appropriate. The introduction can be improved as described above.**

Response: the following changes have been made in the revised Introduction (page 3):

"The broad, pleiotropic working mechanism of hypoxia-mediated metabolic adaptations might have advantages over pharmacotherapeutic approaches which more typically deploys a single-pathway paradigm to achieve disease modification in PD.^{16,17"}

"Contrarily, chronic (intermittent) hypoxia, for example during obstructive sleep apnea, promotes α -synuclein aggregation and mitochondrial dysfunction, and is associated with increased neurodegeneration.^{28,29} This delicate balance between neuroprotection or neurodegeneration²⁹ raises questions about the safety and feasibility as well as the optimal protocol for such strategies in PD."

"However, no studies have systematically investigated the safety, dose-response relation or short-term effects of hypoxic conditioning in PD in a randomized trial.^{30"}

"This trial employed a double-blinded, randomized placebo-controlled multiple N-of-1 design to assess different hypoxic conditioning protocols in all participants using Bayesian and frequentist analyses. This design is especially useful as it supports the dose-finding character of this study, and allows for a randomized intervention order in every individual participant so that participants act as their own control, thus allowing for the comparison of (sub)acute symptom responses across interventions and within individuals."

• **Inflammatory material: There is no inflammatory material.**

Reviewer #2

• **Key results: This paper reports the results of aggregated, randomized, double blinded, placebo controlled N-of-1 trials testing the safety, feasibility (primary outcomes) and symptom improvement (secondary outcome) of using different levels and lengths of hypoxic conditioning on patients with Parkinson's Disease. The authors show the short-term safety and feasibility of the different interventions. They also show that the hypoxic conditioning using the FiO2 level of 0.163 is equal to placebo (normoxic) regarding assessor-reported symptom control, and superior to placebo in patient-reported symptom control. Finally, they showed that the N-of-1 trial design is efficient for performing randomized, controlled trials. The paper can really stimulate further interest, but not without providing more detail on various aspects of the design and analysis.**

Response: we thank both reviewers for their thoughtful comments and suggestions, to which we respond below.

• **Validity: There are a few significant flaws in this manuscript that preclude publication, and they involve too little information provided about the study sample, the design of the study, and the analysis methods. The Results and Methods sections can be improved as described in detailed notes above.**

Response: we have substantially revised the Methods section, paying special attention to the sample size calculation, the complex study design and corresponding statistical analyses, as suggested. A detailed description of our methods was published previously (PMID 35836147, <https://doi.org/10.1186/s12883-022-02770-7>), and we relied on this when we submitted the first version. We now appreciate that this likely resulted in a methods section in our original submission that contained too little detail on the methods. This also refers to some of the comments on the lack of detail listed below. We obviously understand that sufficient detail must be present in the present manuscript to improve readability and providing insight. Specific adjustments are discussed in detail below.

• **Originality and significance: This manuscript is original, both in its use of the aggregated N-of-1 trial design, and in its exploration of controlled hypoxia as a treatment for PD. As such, it holds significance**

for the broad scientific community because it is promoting novel clinical trial design, but also for neuroscientists and neurologists who study and treat PD. Unfortunately, many technical and data analysis details are missing or incomplete and need to be addressed.

Response: we have addressed these concerns by providing more details on the analytical methods and have provided the code including instructions for use. Specific adjustments are discussed in detail below.

• **Data & methodology: The methodology used is not detailed and needs attention. The authors should describe the Bayesian methods used and how they can deal with phenomena associated with N-of-1 trials: 1. Serial or autocorrelation between measurements over time; 2. Carry over effects; 3. Random effects. A mixed model could be used to test and address all these (PMID: 20863658; PMID: 35653405). Also, if some factors, such as BDNF levels are thought to mediate the relationship between hypoxia and clinical outcomes, then mediation analysis should be pursued (PMID: 29283590).**

Response: led by our co-author FG, who is a statistician with special expertise in Bayesian analysis and related models, we now provide explanations for our Bayesian methods and how issues with N-of-1 trials were addressed. Furthermore, we have carried out the recommended mixed model analyses to mitigate these concerns. Specific adjustments are discussed in detail below.

• **Appropriate use of statistics and treatment of uncertainties: All error bars should be defined in the corresponding figure legends; please comment if that's not the case. Please include in your report a specific comment on the appropriateness of any statistical tests, and the accuracy of the description of any error bars and probability values. The authors should provide much more detail about analysis results, parameter estimates, and the statistical significance of the hypothesis tested. Also, they should consider using a correction for multiple comparisons, but acknowledge that in a phase I study links to efficacy are exploratory and can motivate follow-up studies that are appropriately powered based on the phase I results.**

Response: these points are all well taken. We have inspected all effect estimates and corresponding error bars and legends, and have complemented all descriptions where necessary. As suggested by these and other reviewers, we have now modified the frequentist section of our statistical analysis and have provided a rationale for this (see also additional comments below). We now also report the outcome of our results with correction for multiple comparisons, and have added an explicit interpretation of our results to the Discussion. Specific adjustments are discussed in detail below.

• **Conclusions: In current form, the conclusions and data interpretation are not robust, valid or reliable. Hypoxia appears safe, but its broader effects are unclear based on a lack of detail of the analysis and analysis methods.**

• **Suggested improvements: Greater attention to the analysis and analysis methods is needed, as discussed above and in the detailed sections below.**

Response: we have substantially revised the Methods section, paying special attention to sample size calculation, the complex study design and corresponding extra analyses. Specific adjustments are discussed in detail below.

• **References:** Additional references to consider:

- 1) Burtscher, PMID: 37860950 <https://physoc.onlinelibrary.wiley.com/doi/10.1113/JP285230>
- 2) Burtscher, PMID: 33862277 <https://doi.org/10.1016/j.arr.2021.101343>
- 3) Kalva-Filho, PMID: 37872033 DOI: 10.1016/j.parkreldis.2023.105885
- 4) Ho, PMID: 37437730 DOI: 10.1038/s41467-023-39811-9
- 5) Guo, PMID 35959288 DOI: 10.3389/fnagi.2022.919343
- 6) Ferris, PMID: 17414812 DOI: 10.1249/mss.0b013e31802f04c7
- 7) Vermehren-Schmaedick, PMID: 22297041 DOI: 10.1016/j.neuroscience.2012.01.017
- 8) Helan, PMID: 24462831 DOI: 10.1016/j.yjmcc.2014.01.006

Response: thank you for highlighting this important body of work. We have now added all suggested references to the revised manuscript.

• **Clarity and context: The abstract is clear and appropriate. The introduction can be improved as described above.**

Response: the following changes have been made in the revised Introduction (page 3):

"The broad, pleiotropic working mechanism of hypoxia-mediated metabolic adaptations might have advantages over pharmacotherapeutic approaches which more typically deploys a single-pathway paradigm to achieve disease modification in PD.^{16,17"}

"Contrarily, chronic (intermittent) hypoxia, for example during obstructive sleep apnea, promotes α -synuclein aggregation and mitochondrial dysfunction, and is associated with increased neurodegeneration.^{28,29} This delicate balance between neuroprotection or neurodegeneration²⁹ raises questions about the safety and feasibility as well as the optimal protocol for such strategies in PD."

"However, no studies have systematically investigated the safety, dose-response relation or short-term effects of hypoxic conditioning in PD in a randomized trial.^{30"}

"This trial employed a double-blinded, randomized placebo-controlled multiple N-of-1 design to assess different hypoxic conditioning protocols in all participants using Bayesian and frequentist analyses. This design is especially useful as it supports the dose-finding character of this study, and allows for a randomized intervention order in every individual participant so that participants act as their own control, thus allowing for the comparison of (sub)acute symptom responses across interventions and within individuals."

• **Inflammatory material: There is no inflammatory material.**

SPECIFIC COMMENTS

Introduction

- **This section could be improved. It would benefit from a discussion of the dual role of hypoxia as either causing neurodegenerative processes such as PD or allowing for neuroprotection. Currently, the authors only discuss the neuroprotection possibilities from hypoxia. The following recent references can be added to stress the point that hypoxia is a double-edged sword with PD:**

- **Hypoxia leads to increased aggregation of the protein alpha-synuclein, which has been implicated in Lewy body formation in PD – (PMID: 37860950, 35959288)**
- **Hypoxia leads to increased mitochondrial dysfunction, which is a hallmark of PD – (PMID: 37433770)**
- **This review article addresses this issue (PMID: 33862277)**

- **By laying out this dichotomy in the introduction, a stronger argument is made for the author's Phase 1 study to determine the appropriate dose-response of hypoxic conditioning for PD.**

Response: we appreciate this important suggestion and have revised the Introduction, which now reflects this delicate balance between neuroprotection and neurodegeneration and cites the recommended relevant articles.

Manuscript changes (p. 3):

"The broad, pleiotropic working mechanism of hypoxia-mediated metabolic adaptations might have advantages over pharmacotherapeutic approaches which more typically deploys a single-pathway paradigm to achieve disease modification in PD.^{16,17} Targeting the hypoxia response pathway might be a promising novel treatment strategy in PD with the potential to alleviate symptoms, and which might ultimately modify the course of PD.¹¹ Hypoxic conditioning has been used in a variety of populations, including individuals with spinal cord injury and multimorbidity, without notable adverse effects¹⁸⁻²⁷. Contrarily, chronic (intermittent) hypoxia, for example during obstructive sleep apnea, promotes α -synuclein aggregation and mitochondrial dysfunction, and is associated with increased neurodegeneration.^{28,29} This delicate balance between neuroprotection and neurodegeneration²⁹ raises questions about the safety and feasibility and the optimal protocol for such strategies in PD. However, no studies have systematically investigated the safety, dose-response relation or short-term effects of hypoxic conditioning in PD in a randomized trial.^{30"}

- **Also, there should be more background on the nature of the aggregated N-of-1 design used and why it makes sense. This includes the ability to randomize the order of the interventions for any one patient given the short time it takes to administer them (the length of time it takes to administer the hypoxic challenge should be discussed as well) as well as the timing of hypoxia's effects; i.e., they must be acute, but how lingering are they? How long do the effects of any one hypoxic challenge last in terms of their further effects on clinical endpoints? In this context the authors should consider the possibility of carryover effects in conducting short term N-of-1 trials that have adjacent time periods in which different hypoxic challenges are administered.**

Response: in both the revised Introduction and Methods section, we now more explicitly explain the rationale for using this aggregated N-of-1 approach. Furthermore, we have provided references for the timing of interventions in this trial and have clarified the timeline of the protocol in *Figure 1*. This Figure details the order of events in every week during which an intervention is organized. To evaluate the lingering of (sub)acute effects, participant-reported symptoms were rated for three days after every intervention, despite the first window of conditioning effects being up to one day. Indeed, we recognize that after a day, effects on participant-rated symptoms subside. We have implemented all these changes in the methods, results and discussion section.

Manuscript changes (Introduction, p. 3):

"In this phase 1 trial, we assessed the safety, feasibility, and short-term effects of different individual-session protocols of hypoxic conditioning in individuals with PD. This trial employed a double-blinded, randomized placebo-controlled multiple N-of-1 design to assess different hypoxic conditioning protocols in all participants using Bayesian and frequentist analyses. This design is especially useful as it supports the dose-finding character of this study, allows for a randomized intervention order in every individual participant and participants act as their own control, thus allowing for the comparison of (sub)acute symptom responses across interventions and within individuals."

Methods (p. 12-13):

"The Bayesian analysis allowed for an estimation of the posterior probability that any specific intervention protocol results in a clinically beneficial effect at both group and individual level. In addition, these methods are useful in slowly progressive disease and in exploratory research phases.^{72,100} The measure upon which secondary outcomes were compared across different intervention protocols in the Bayesian analysis was defined as the standardized area under the curve of the consecutive measurements (scores) from 30 minutes up until 1 day post intervention. These scores were calculated for the placebo and the 4 active intervention protocols. Specifically, we set a hierarchical model to account for both between-patient variability (through patient-level means) and within patient measurement error. This hierarchical structure borrows strength across patients to better estimate overall treatment effects. We assumed that outcomes followed a normal distribution with individual treatment means for the intervention protocols around overall intervention group means. Vague priors were assigned to the means, and uniform priors over a large range were defined for the standard deviations. We then computed the overall difference for each intervention protocol and the placebo, but also for each individual. Importantly, we derived the posterior probability of reaching the predefined MCID of 0.75 points between baseline and post-intervention. If this posterior probability was greater than 80%, interventions were considered effective. Interventions with posterior probabilities <20% were considered ineffective."

Discussion (p. 7):

"Although this dosage falls in the range of previous dose-finding studies that demonstrated a favorable benefit-harm ratios^{39,40}, most hypoxic conditioning studies have deployed F_{iO_2} in lower ranges.⁴¹ In animal studies, a short-lived but strong dose-response relation between F_{iO_2} and striatal dopamine release was observed, suggesting a potential mediating mechanism for short-term effects.⁴ In our study, we could not demonstrate such a dose-response relation in either symptoms or serum markers for target engagement. The observed duration of symptom improvement was in line with the first window of hypoxic conditioning, which suggests that effects subside after several hours.⁴² It should be noted that we only investigated single hypoxic conditioning sessions and two different F_{iO_2} levels."

Figure 1 is now clarified (p. 22):

"Figure 1: Study protocol for every participant per week. As an example, the contents of week 1 are highlighted. All subsequent intervention weeks are identical in events and measurements."

Methods:

- Please give more detail regarding how you chose the sample size for this study. You state in the study population that the 'sample size was informed by a previous multiple N-of-1 trial experience' and cite the JAMA article, but this gives readers no real understanding of why you chose 20. Does this mean you just decided that since the JAMA article on mexiletine had 30 individuals, you would have 20?

Response: As the primary outcomes of this study were safety and feasibility, a formal power calculation for the secondary or exploratory outcomes were not previously part of the originally submitted manuscript. However, in answer to your question, we have performed a post-hoc sample size calculation, which was informed by our previous experience and assumptions based on the minimal clinically important difference for our participant-rated secondary outcomes. We have now added this clarification to the revised manuscript.

Discussion (p. 8):

We conducted a post-hoc sample size calculation for a future randomized control trial with intermittent hypoxia at F_{iO_2} 0.163 and a control/placebo group (calculation publicly accessible at <https://github.com/Federica-Giardina/Talisman>). We first estimated the effect and precision using a mixed model adjusting for confounders (details below, Statistical Analysis section). For the precision, each individual contributed the average of 2 repeated measurements. A two-sample t-test with a significance level of 0.05 and 80% was used. This reveals that with observed effects on participant-reported outcomes, a sample size of $n=45$ (for the participant-selected prime symptom) or $n=158$ (for urge to take dopaminergic medication) per treatment group would be required in a future randomized-controlled trial. Lastly, our study focused on the immediate effects of single exposure to hypoxic conditioning. Therefore, future clinical and translational studies should focus on the clinical effects with higher-frequency and multiple-week interventions and mechanisms of hypoxic conditioning in PD, as well as explore potential small-molecule approaches towards hypoxia response cascade activation in PD.

- For the trial design and execution, there are quite a few things that are unclear. For example, how long was each intervention? 30-45 minutes? Were they all done on a single day or 1 on each of 5 different days over two weeks in which the 5 interventions were administered twice? Were the second week's interventions also randomized independently of the first week's? Since the interventions were given in a random order to each patient, the authors should simply make a figure with the basic layout in hours or days, and what went on for each patient in terms of the order of the interventions (e.g., 20 patients correspond to 20 rows in the table (with a header row with a title for each column), with each column reflecting the order of the 10 interventions where each patient's cells in the table would contain what intervention they were provided in that time).

Response: we have clarified the study protocol in our revised manuscript. Although a detailed protocol paper on our study was published previously, we appreciate this reviewer's comments and have now clarified several elements of our methods. Briefly, participants visited the hospital *ten* times to receive an intervention. All measurements were conducted during every intervention day, as noted in the new *Figure 1*. Interventions took place across ten consecutive weeks, with a wash-out period between any two interventions of at least five days. This assures minimization of any possible effect lingering. Moreover, potential carry-over effects were minimized by randomizing the order of interventions between individuals. For most participants, this protocol with weekly interventions consisted of ten weeks (occasionally slightly longer, e.g. due to a vacation in-between sessions). A Latin square design was adapted for randomization, and randomization was indeed conducted in two separate blocks for the first and second half of the study protocol. All interventions lasted 45 minutes.

Manuscript changes:

“Every participant received two sets of five different conditions in randomized order, consisting of four active interventions and one placebo, all with a 45-minute total duration. Per week, one intervention was administered, adding up to a total intervention phase of 10 weeks:

- *Continuous hypoxia at FiO₂ 0.127 or 0.133.*
- *Intermittent hypoxia with 5x5 minutes at FiO₂ 0.127 or 0.133, interspersed with 5 minutes normoxic recovery.*
- *Continuous hypoxia at FiO₂ 0.163.*
- *Intermittent hypoxia with 5x5 minutes at FiO₂ 0.163, interspersed with 5 minutes normoxic recovery.*
- *Continuous normoxia (placebo)”*

“Every week across ten consecutive weeks, participants visited the hospital to receive one of the 5 possible interventions (Figure 1). Period and carry-over effects were mitigated by the long wash-out period between interventions, as well as by implementing a new baseline measurement before the start of every intervention. This measurement was then used to calculate subsequent short-term symptomatic effects of that intervention.”

“Participants were equally divided in five groups with different interventions sequence according to a Latin square design with five ‘periods’ of interventions. These periods were grouped in two separate sets, each consisting of a randomized sequence with five periods (Figure 1). Randomization was conducted by a biostatistician (FG) using the R programming language.”

- I don't understand the Bayesian analysis and what it entails, so that should be described in much

greater detail. Also, carryover effects as well as serial correlation and random effects should be explored (see comments below in the analysis subsection of the results section)

Response: we have now made the explanation of the Bayesian methods detailed and have made the code of the analysis available, enabling replication of our results. We have conducted the suggested additional exploratory analyses for serial correlation for the MDS-UPDRS part III (as we have not repeated these measurements beyond the expected effective time window, as we have with participant-rated symptoms), and demonstrate no such effects.

Manuscript changes:

Results (p. 6):

There was no difference in MDS-UDPRS measurements over time in random effects mixed models in participants or on group level ($\beta = 0.016$, $P = 0.88$), indicating no signs of serial correlation.

Methods, Interventions (p. 10):

Period and carry-over effects were mitigated by the long wash-out period between interventions as well as implementing a new baseline measurement before the start of every intervention, which is then used to calculate short-term symptomatic effects of that intervention.

Methods, Outcome measures (p. 11):

“Acute effects on PD symptoms were analyzed by participant-reported and assessor-rated standardized motor scales that were measured 30 minutes pre- and 30 minutes post-intervention, unless indicated otherwise. This means that before every intervention, a novel baseline measurement of secondary and explorative outcomes was conducted.”

Methods, Statistical analysis (p. 11-12):

In accordance with our earlier aggregated N-of-1 experience⁶⁶, secondary outcomes were analyzed using linear mixed models as well as Bayesian analyses by aggregating 20 N-of-1 trials totaling 200 interventions. All secondary outcomes post-intervention were compared to the unique baseline measurements before every intervention to reduce period effects. We fitted a linear mixed-effects model to analyze secondary outcomes across 20 patients, each observed over 7 time points and repeated over 2 sets. One individual had a missing measurement at one day follow-up for participant-rated scales, which was interpolated taking the average of the measurement directly before and the measurement directly after. The model includes 5 interventions, with intervention 5 (placebo) set as the reference. Fixed effects included intercept, time, interventions main effects and treatment-by-time interactions. The model adjusted for BDNF, gender and disease severity. P values are 2-sided, and $P < 0.05$ was the significance threshold. We included a random slope for time at the patient level, allowing the time effect to vary across individuals. This accounts for patient-specific variability in the change of outcomes over time.

The Bayesian analysis allowed for an estimation of the posterior probability that any specific intervention protocol results in a clinically beneficial effect at both group and individual level. In addition, these methods are useful in slowly progressive disease and in exploratory research phases.^{66,95} The measure upon which secondary outcomes were compared across different intervention protocols in the Bayesian analysis was defined as the standardized area under the curve of the consecutive measurements (scores) from 30 minutes up until 1 day post intervention. These scores were calculated for the placebo and the 4 active intervention protocols. Specifically, we set a hierarchical model to account for both between-patient variability (through patient-level means) and within patient measurement error. This hierarchical structure borrows strength across patients to better estimate overall treatment effects. We assumed that outcomes followed a Normal distribution with individual treatment means for the intervention protocols around overall intervention group means. Vague priors were assigned to the means, and uniform priors over a large range were defined for the standard deviations. We then computed the overall difference for each intervention protocol and the placebo, but also for each individual. Importantly, we derived the posterior probability of reaching the predefined MCID of 0.75 points between baseline and post-intervention. If this posterior probability was greater than 80%, interventions were considered effective. Interventions with posterior probabilities <20% were considered ineffective.

- Also, you state the study population was ‘without cardiorespiratory comorbidity’ but this is too vague. Did you exclude people with any diagnosis of coronary artery disease? Previous cerebrovascular accident or myocardial infarction? Arrhythmia? Confirmed diagnosis of asthma or COPD of any degree or just based on use of differing strengths of inhalers or inhaled steroids? Smokers vs non-smokers? Etc., etc. Please provide more details on these and any other important covariates or, if these data weren’t gathered, explain why they are missing from your study population.

Response: we excluded all individuals with a confirmed current diagnosis of asthma or COPD (independent of current treatment), as well as individuals with abnormal pulmonary function testing (obstructive, restrictive or diffusion deficits). Furthermore, we excluded individuals with congestive heart failure and cardiac arrhythmia, including atrial fibrillation. Current smokers were excluded, and former smokers were included only if they had

normal pulmonary function tests. We have now clarified this in the revised Methods section and Supplementary Materials and have made available extensive demographic characteristics in *Table 1*.

Manuscript changes (p. 8):

“Twenty participants with clinically diagnosed PD (Hoehn & Yahr stage 1.5-3) without current cardiorespiratory comorbidity and deep brain stimulation, were included from a national PD research recruitment database. Other inclusion and exclusion criteria are summarized in Supplementary Materials.”

- Expanded Table 1 containing a detailed description of demographic characteristics.
- Added a new table: *Supplementary Table 1* (p. 23) with all inclusion and exclusion criteria.

- Did you consider measuring serum brain-derived neurotrophic factor (BDNF) as an exploratory outcome? BDNF levels are known to be decreased in PD and have been shown to increase in response to controlled hypoxia. (PMID: 17414812, 2229704, 24462831). If one MOA of hypoxia is to affect BDNF levels, then the proper analysis to link hypoxia to symptom improvement is to consider BDNF as a mediator in the analysis (see, e.g., PMID: 29283590)

Response: we have indeed considered this. We had previously decided to not include an analysis of serum BDNF, as we expected limited effects due to the short timeframe of the outcome measurement in this study. However, as some clinical and preclinical studies indeed suggest a possible short-term effect on BDNF, we have now included serum BDNF as an additional mechanistic outcome measure. This has now been added to the Results and Discussion. The complete analyses are reported in the Supplementary Materials.

Manuscript changes (Results, p. 5):

“Adding BDNF as an interaction term further strengthened the post-intervention improvement on self-selected symptoms (0.68, CI 95% 0.32-1.04), but weakened the improvement on urge to take dopaminergic medication (0.40, CI 95% -0.002–0.80). BDNF response did not affect global symptom results (0.012, 95% CI -0.023–0.00).”

Methods: “As an additional exploratory marker of neuroprotection, analysis of BDNF was included. Preclinical and clinical studies suggest acute effects of hypoxia on BDNF^{83,84} and BDNF levels are associated with symptom severity in PD, possibly reflecting reduced neuroplasticity.^{85,86}

[...]

As explorative analyses, BDNF, gender and disease severity were added as covariates.”

Results:

- For Table 1, Demographic characteristics, please provide a more robust table with further details regarding characteristics that would be of particular importance to a study on hypoxic conditioning, as described in the comments above on Methods. Also include known use of any medications or deep brain stimulators for Parkinson’s Disease.

Response: we have significantly expanded the table to now provide a more detailed characteristic of the study population, including detailed information regarding medication use. An active deep brain stimulation system was a contra-indication in this study, as we regarded the frequent (i.e., weekly) OFF-medication assessments too burdensome for this subgroup of patients.

Manuscript changes (Methods, p. 9):

“Twenty participants with clinically diagnosed PD (Hoehn & Yahr stage 1.5-3) without current cardiorespiratory comorbidity and without deep brain stimulation were included from a national PD research recruitment database. Other inclusion and exclusion criteria are summarized in Supplementary Materials.”

- Expanded Table 1 containing a detailed description of demographic characteristics.

- How often was blood drawn? It is unclear in, e.g., Supplementary figure 12 exploring cortisol levels.

Response: during each of the 200 testing days, blood was drawn at five timepoints to allow for longitudinal measurements of cortisol throughout the morning. All other analyses (EPO, PDGFR β , BDNF, NfL, GFAP) were based on one sample 30 minutes pre-intervention and compared to a sample taken 60 minutes post-intervention. This is now better explained in the revised Methods.

Manuscript changes (Methods, p. 11):

“Briefly, blood was drawn using venipuncture in a serum tube at five different timepoints: two times before intervention, separated by 30 minutes; one time directly after intervention; and then after 30 and 60 minutes post-intervention. After refrigeration for 30 to 60 minutes, serum was centrifuged for 10 minutes at 2000 rcf, 4°C. Supernatant serum was collected and stored at -80°C until further analysis. For all analyses except cortisol, only the samples 30 minutes pre-intervention and 60 minutes post-intervention were tested.”

“Supplementary Figure 11: LMM of PDGFR β (A), EPO (B), cortisol (C), GFAP (D), NfL (E) and BDNF (F). For all markers, blood was drawn 30 minutes pre-intervention and 60 minutes post-intervention. An exception is cortisol, for which extra in-between measures were taken directly pre-intervention, directly after intervention and 30 minutes after the intervention to account for circadian rhythm effects.”

Discussion:

- Since there were no significant effects of hypoxia that would stand up to multiple comparisons adjustments, the authors should provide guidance in the discussion section on how large a study would be needed to adequately test the hypothesis of positive hypoxia effects based on the observed effect size in the N-of-1 trials. Also, the authors should address the clinical relevance of any assumed effect size.

Response: we agree with the reviewer, and have conducted a sample size calculation given the established effects in this current study to inform any subsequent (two-armed) protocols. The clinical relevance of established effects in this study was defined as 0.75 points on the MCID of 10-point Likert scales, and are pre-defined for the other assessor-rated items (e.g., MDS-UPDRS part III). These are taken into account in the sample size calculations.

Discussion (p. 6-7):

“Finally, the intermittent hypoxia protocol at F_IO₂ 0.163 seemed most promising compared to the other hypoxia protocols for short-term improvement in participant-rated symptom scales, although the effects were modest. Also, assessor-rated assessment did not show significant acute improvement.”

[...]

“Our secondary aim related to the effects of hypoxic conditioning on clinical outcomes, and we found that participant-rated symptom scales suggest modest short-term symptomatic effects of intermittent hypoxia at F_IO₂ 0.163 relative to placebo and other conditioning protocols, although the effects were not consistent across outcome measures.”

Discussion (p. 8):

“We conducted a post-hoc sample size calculation for a future randomized control trial with intermittent hypoxia at F_IO₂ 0.163 and a control/placebo group (calculation publicly accessible at <https://github.com/Federica-Giardina/Talisman>). We first estimated the effect and precision using a mixed model adjusting for confounders (details below, Statistical Analysis section). For the precision, each individual contributed the average of 2 repeated measurements. A two-sample t-test with a significance level of 0.05 and 80% was used. This reveals that with observed effects on participant-reported outcomes, a sample size of n=45 (for the participant-selected prime symptom) or n=158 (for urge to take dopaminergic medication) per treatment group would be required in a future randomized-controlled trial.”

Discussion (p. 9):

“The established effects on participant-rated outcomes were modest, yet clinically relevant, and warrant further investigation because of the low cost and burden of this non-pharmacological intervention.”

Reviewer #3

The present manuscript by Janssen Daalen et al. reported some novel findings from a Phase 1 trial investigating the effects of hypoxic conditioning on symptoms and serum biomarkers in 20 patients with Parkinson’s Disease (PD). Overall, to my knowledge, this is the first randomized, double-blinded, placebo-controlled N-of-1 trial on a promising non-pharmacological intervention (i.e. controlled systemic intermittent hypoxia) for symptom control and disease modifying in PD patients. The apparent strengths of this group of authors include their expertise in clinical management of PD and design of N-of-1 trials (e.g. Ref. 56, JAMA 2018). In brief, this study represents a major advance to bring a largely experimental approach – hypoxic conditioning into a clinical reality in alleviating such a severe neuro-motor disorder – PD, which remains incurable with the current pharmacotherapy. Nevertheless, I have the following few concerns or suggestions for the authors to consider for further improvement of the manuscript.

1) Above all, while the goals in Primary Outcomes have been well achieved, these results in safety and adverse events are somewhat predictable, based on numerous previous clinical studies in various healthy or patient subjects. In addition, in 3rd Outcomes measures, two selected biomarkers – EPO and PDGFR β were largely unaffected by the single exposure to intermittent hypoxia. Could the authors perform additional analyses that may result in more mechanistic insights that will stimulate further investigation. For example, could you further check if other proposed biomarkers of PD could be modified by various regimens of hypoxic conditioning? Please consider the following biomarkers of interest: hypoxanthine (a purine metabolite altered by PD, see PMID: 32903216); neurofilament light (NFL) and glial fibrillary acid protein (GFAP) see PMID: 36997567; and C-reactive protein (see PMID: 31693246). Any

positive findings in these new biomarkers will greatly enhance this work in terms of extraordinary innovation and quality required by Nature Communications.

Response: for the current study, we had initially decided to not include intermediary markers such as EPO or PDGFR β , as we considered that the study's exposure to hypoxia interventions might not have a relevant impact on these neuronal damage markers. In retrospect, however, recent evidence suggests that BDNF might be upregulated by even a single hypoxic exposure, with a similar time interval to ours, and NfL and GFAP might give insight into whether acute effects evoke neuronal stress signals. Therefore, we have now conducted additional analyses of BDNF, NfL and GFAP for all interventions. These analyses have now been added to the revised *Methods, Results and Discussions*.

Manuscript changes (Results, p. 6):

"Soluble platelet-derived growth factor receptor beta (PDGFR β), erythropoietin (EPO), neurofilament light-chain (NfL) and brain-derived neurotrophic factor (BDNF) were not changed one hour post-intervention (linear mixed models, $P > 0.05$ for all analyses). Glial fibrillary acidic protein (GFAP) increased across all groups (mean 29.6 pg/mL, $P < 0.001$), but there were no differences between subgroups. Apart from a significant circadian rhythm-mediated decrease in cortisol during all interventions (0.37 to 0.24 $\mu\text{mol/L}$, $P < 0.01$), there were no between-intervention differences ($P > 0.31$, Supplementary Materials)."

Discussion (p. 8):

"GFAP and NfL, as markers of neuronal stress and neural degeneration and potential biomarkers of PD progression^{65,66}, were not altered after single sessions of hypoxic conditioning, reflecting no acute neuronal injury within two hours after onset of hypoxia. It should be noted that the first studies with these novel biomarkers suggest that in traumatic brain injury, increase in serum becomes apparent only several hours post-injury^{67,68}. Future studies should examine whether these markers are affected after longer hypoxic conditioning protocols. On the other hand, activation of the hypoxia response pathway is linked to several PD risk genes, including LRRK2, DJ-1 and PINK1-Parkin.⁴⁹ Therefore, activation of using specific hypoxic conditioning protocols might have protective or compensatory properties. Future studies could investigate the effects of hypoxic conditioning on mitochondrial function and oxidative stress, such as by studying peripheral blood mononuclear cells (PBMCs)."

Methods (p. 11-12):

"As exploratory serum markers of target engagement, dose-response effects and induced mechanisms, EPO and PDGFR β were measured in serum pre-intervention and compared to 60 minutes post-intervention. EPO is strongly activated in response to hypoxia in healthy controls^{43,54} and less strongly in elderly^{43,85-87}, but response in people with PD is unknown. PDGFR β is a tyrosine-kinase receptor that is shed by brain pericytes in response to hypoxia⁸⁸. Transient activation of its signaling pathway is associated with neuroprotective effects through PI3k/Akt activation^{52,59,88}. As an additional exploratory marker of neuroprotection, analysis of BDNF was included. Preclinical and clinical studies suggest acute effects of hypoxia on BDNF^{89,90} and BDNF levels are associated with symptom severity in PD, possibly reflecting reduced neuroplasticity.^{91,92} Serum NfL and GFAP were included as acute markers of neuronal injury.⁹³ NfL and GFAP are both associated with symptom severity in PD.^{68,94-96} Early evidence shows that exercise, a conditioning intervention with overlapping working mechanisms compared to hypoxic conditioning¹⁵, reduces NfL.⁹⁷ Notably, both NfL and GFAP are induced by chronic intermittent hypoxia, such as occurring in sleep apnea.^{98,99} These markers were analyzed using ultrasensitive single-molecule array (Simoa, Neurology 2-PLEX, Quanterix[®], USA)."

2) The style and quality of Tables and Figures are not very impressive. For example, Table 1 appears oversimplified. More morphometric and medical information should be added to Table 1, for instance, racial background, BMI, medical history, including medication taken, etc.

Response: Thank you for pointing this out. We have now made this Table more comprehensive by including the mentioned parameters, as well as various baseline levels of assessor-rated items, Parkinson's disease questionnaire-39 and PD- and non-PD-related medication. Furthermore, we have upgraded and homogenized the visualization of results across all figures, including the figures' styles as suggested by the reviewers, including the figures in the Supplementary Materials.

Manuscript changes (p. 21):

- **Table 1** expanded with detailed demographic characteristics
- Tables throughout Supplementary Materials updated and refined in high-quality

3) The current in-text Table 1 and Figure 1 are not impressive. Do you think inclusion of Supplemental Figure 7 and/or Figure 8 as in-text figure(s) would be more appealing?

Response: we agree with the reviewer and have now integrated Supplementary Figure 8 (containing the three participant-reported symptoms) with the main text as Figure 4. Furthermore, Table 1 has been expanded and now includes a comprehensive baseline description. Figure 1 now contains a visualization Study Protocol, Figure 2 shows the participant flowchart and Figure 3 displays the categorization of adverse events.

Manuscript changes:

- Modified Figure 1 as suggested by the reviewer (p. 21)
- Added a participant flow chart as Figure 2 (p. 21)
- Expanded Table 1 with detailed demographic characteristics (p. 23)
- Figure 4 modified and added to main text (p. 23-25):
“**Figure 4.** Change in the participant-selected symptom (usually the most prominent parkinsonian symptom, A), urge to take dopaminergic medication (B) and global symptom impression (C). These were rated on a 10-point Likert scale pre- and post-intervention. The delta between pre-intervention and all post-intervention assessments is depicted below. 0 minutes is the first post-intervention assessment, taken directly after the intervention.”

4) I would suggest using “Violin plots” to present the data instead of the current bar graphs in Supplementary Figure 3, 4, 5, 6. Violin plots also show each individual data points and improve data transparency.

Response: We thank the reviewer for this excellent suggestion, and accordingly we have modified the figures to violin plots.

Manuscript changes (p. 24-27):

These figures (now numbered Figures 4-5-6-7) are now depicted as violin plots.

5) In Supplementary Figure 11, please choose different types of symbols for each of the treatment conditions. This will help readers visually to differentiate the treatment conditions.

Response: we have modified this Figure as well as Supplementary Figure 12 to help with visual interpretation.

6) Adding a new figure to provide an illustrative description of the study protocol, group assignment and timelines in detail would help the readers’ comprehension.

Response: we have implemented this Figure into the main text to help with comprehension of the study protocol.

7) Although both male and female PD patients 50/50 participated in this study, the authors did present nor discuss any gender-dependent differences in response to hypoxic conditioning.

Response: as part of the newly added linear mixed model analysis, we now report a post-hoc analysis of between-gender differences in response to all secondary and exploratory (serum) outcomes. Although speculative, we have added a brief interpretation of this to the discussion.

Manuscript changes (Results, p. 5):

“Women reported significantly more symptomatic improvement on all three participant-rated symptoms compared to men (estimates between 0.39 and 0.90). Disease severity did not modify the observed effects.”

Discussion, p. 7:

“Interestingly, women in our cohort reported more positive results on all participant-rated outcomes compared to men. Women are generally underrepresented in PD trials and a better understanding of gender differences is imperative. Factors to possibly consider include the notion that striatal binding on DAT scans is generally higher in women compared to men throughout the disease course, and also that levodopa-induced dyskinesias are more common in women (acknowledging that this could result directly from relative overdosing in women)^{45,46}. Although speculative, it is possible that a gender-related difference in striatal sensitivity to the short-term effects of hypoxic conditioning might explain some of the observed effect differences across genders. Future hypoxic conditioning studies should further investigate these effect differences.”

8) Numerous errors can be seen in the References. Please carefully proofread and correct the references.

Response: we have carefully reviewed all References and have corrected the errors.

9) A full description of each of the abbreviated terms should be done at their first appearance in the text, not at the end of the paper (Methods).

Response: we have carefully checked all abbreviations and have made sure that these are fully described in their first appearance in the manuscript.

Reviewer #4

This manuscript reports a preliminary clinical trial testing the safety and feasibility of continuous and intermittent exposures to mild-moderate hypoxia, administered over 45 minutes, in Parkinson's Disease patients. The hypoxia exposures were well tolerated and were not associated with increased incidence or severity of adverse events, compared with normoxia placebo. A strength of this study is the repeated measures N-of-1 design, in which each subject completed two sets of all 5 exposures (normoxia, 45-min continuous mild or moderate hypoxia, and 45-min intermittent mild or moderate hypoxia). The hypoxia sessions did not produce statistically significant improvements in the subjects' Parkinson's Disease symptoms, which is not surprising given that clinical studies of intermittent hypoxia intervention for heart failure [Saeed et al., J Card Fail 2012;18:387-91; PMID 22555269], prediabetes [Serebrovska et al., High Alt Med Biol 2019;20:383-91; PMID 31589074] and cognitive impairment [Wang et al., Am J Alzheimers Dis Other Demen 2020;35:1533317519896725] showed multiple IH sessions/week for several weeks were required for appreciable benefits. Nevertheless, this study establishes safety of intermittent hypoxia therapy in PD patients.

Thank you for your careful assessment of our paper, and for highlighting these important references, which are now addressed in the revised manuscript.

Major Comments

Was a power analysis done a priori, to determine if 20 subjects would provide sufficient statistical power (i.e. $1-\beta$)?

Response: as our primary outcomes were safety and feasibility, we did not previously perform a sample size calculation on these outcomes. However, we have performed a post-hoc sample size calculation, which will also inform sample size for our phase 3 follow-up trial, given the effect observed in the current study.

Discussion (p. 8):

"We conducted a post-hoc sample size calculation for a future randomized control trial with intermittent hypoxia at F_{iO_2} 0.163 and a control/placebo group (calculation publicly accessible at <https://github.com/Federica-Giardina/Talisman>). We first estimated the effect and precision using a mixed model adjusting for confounders (details below, Statistical Analysis section). For the precision, each individual contributed the average of 2 repeated measurements. A two-sample t-test with a significance level of 0.05 and 80% was used. This reveals that with observed effects on participant-reported outcomes, a sample size of $n=45$ (for the participant-selected prime symptom) or $n=158$ (for urge to take dopaminergic medication) per treatment group would be required in a future randomized-controlled trial."

Please explain your rationale for applying 45 minutes of continuous hypoxia, rather than 25 minutes, which would match the hypoxia "dose" (i.e., 5 x 5 minutes) of the intermittent hypoxia protocol.

Response: although we agree that the total hypoxia dosage would be more similar if continuous hypoxia were administered for 25 minutes, we regard the recovery (normoxic) phase during intermittent hypoxia as an important and active part of the intervention itself. Furthermore, we were primarily interested in the potency of an intermittent stimulus and the trade-off this has for the potency of a short-term effect on symptoms and exploratory outcome markers. This is now clarified in the revised manuscript.

Manuscript changes (p. 9):

"Although the total hypoxic dosage in the intermittent hypoxia interventions is lower due to the in-between normoxic bouts, we were interested in comparing same-duration interventions in this dose-finding study and ensuring all other variables remain constant across interventions."

Please include body mass index (BMI) in Table 1. The BMI range of the participants may be of interest since a recent meta-analysis revealed an association of Parkinson's disease with low BMI [Li et al., J Neurophysiol 2024;131:311-20; PMID 38264801].

Response: we have now made this Table more comprehensive by including BMI, as well as various baseline levels of assessor-rated items, Parkinson's disease questionnaire-39 and PD- and non-PD-related medication.

Minor Comments

Several acronyms appear in the manuscript and supplemental materials. To assist the reader, nonstandard abbreviations must be defined where they first appear. Several (SaO₂, FiO₂, MDS-UPDRS, MCID, EPO) are defined in the Methods, which follows the other manuscript sections, and "TUGT" is

defined in the supplemental materials. Please move these definitions to the first uses of the abbreviations. ON, OFF, PPT and LMM are not defined; please do so.
Abstract: Delete the 9th sentence (“20 participants...”); the 3rd sentence (“20 individuals with PD...”) already presents that information.

Stop criteria subsection of Results: Delete “intervention” after screening, since screening is not an intervention.

Screening Procedure paragraph (Methods): The first sentence indicates spirometry is done by measuring CO diffusion capacity, but it isn't. Did you mean to say “...using spirometry and by conducting a carbon monoxide diffusion capacity test.”?

Supplemental Table 1: Should the legend read “Green indicates greater or equal to...”?

Response: we thank the reviewer for the alertness to these descriptions. This is now corrected, and all recommended adjustments have been implemented.

Supplemental Table 2: “during screening” not “during screening day”

Response: this has now been corrected.

Supplemental Figures 1-11: Please indicate if the vertical bars represent SD, or SEM.

Response: these refer to standard deviations, which has now been added to all figure descriptions.

Supplemental Figure 2: Were the hypoxia exposures continuous, or intermittent?

Response: In order to evaluate the autonomic responses, we previously only included continuous hypoxia interventions. However, we have now included the autonomic responses to all five interventions, including intermittent hypoxia, to demonstrate whether autonomic response fluctuates with normoxic and hypoxic intervals. We trust this gives better insight in safety-related aspects of such interventions.

Manuscript changes (p. 26):

Addition of Supplementary Figures 2C and 2D.

Supplemental Figures 4-7: The Figure numbers in the legends are incorrect.

Response: these were indeed superfluous and have now been removed.

Reviewer #5

This is an interesting work by Professor Bloem and colleagues, studying the effect of hypoxic conditioning in patients with Parkinson's disease (PD), in a phase 1, randomized controlled multiple N-of-1 trials. The intervention is novel in PD, as well as the trial design. The primary endpoint of the trial is to assess safety, feasibility and short-term effects of different protocols of hypoxia in PD. Each participant completed 4 sessions of different hypoxia protocols (two different levels of FiO₂, either intermittent or continuous) and one placebo, once weekly, and this was repeated twice. The order of the sessions differed among participants.

Please find here my comments and questions to the authors:

Introduction

- The first paragraph and the section “In addition, converging evidence suggests that repeat exposure to moderate hypoxia induces an evolutionary conserved adaptive survival mechanism, termed hypoxic conditioning.

Adaptive responses involve improved cellular energy metabolism, which in PD is disturbed by mitochondrial dysfunction, a subsequent reduction in oxidative stress and induction of adaptive plasticity” miss references.

Response: references to these statements have now been added.

Methodology

- **The selection and profile of participants is a bit unclear. Although mentioned in the previously published open-access protocol I think it would be informative to state the inclusion and exclusion criteria.**

Response: during screening, detailed history-taking was performed by a medical doctor, including recent smoking and cardiovascular abnormalities, most importantly arrhythmias and congestive heart failure. These were an exclusion criterion for this first hypoxic conditioning study in individuals with PD, and these are now also listed in the new *Supplementary Table 1*. We now elaborate in greater detail on all inclusion and exclusion criteria in the revised Methods section, *Supplementary Table 1* and describe in *Table 1* demographic characteristics of our participants. We have also performed additional post-hoc analyses on subtypes, such as gender and disease severity, and reflect on these in the Discussion.

Manuscript changes (p. 23):

- **Supplementary Table 1:** complete list of inclusion and exclusion criteria added
- **Table 1** expanded with detailed demographic characteristics

Results (p. 5):

“Women reported significantly more symptomatic improvement on all three participant-rated symptoms compared to men (estimates between 0.39 and 0.90). Disease severity did not modify the observed effects. Adding BDNF as an interaction term further strengthened the post-intervention improvement on self-selected symptoms (0.68, 95% CI 0.32-1.04), but weakened the improvement on urge to take dopaminergic medication (0.40, 95% CI 0.00–0.80). BDNF response did not affect global symptom results (0.012, 95% CI -0.023–0.00).”

Discussion (p. 7):

“Interestingly, women in our cohort reported more positive results on all participant-rated outcomes compared to men. Women are generally underrepresented in PD trials and a better understanding of gender differences is imperative. Factors to possibly consider include the notion that striatal binding on DAT scans is generally higher in women compared to men throughout the disease course, and also that levodopa-induced dyskinesias are more common in women (acknowledging that this could result directly from relative overdosing in women)^{46,47}. Although speculative, it is possible that a gender-related difference in striatal sensitivity to the short-term effects of hypoxic conditioning might explain some of the observed effect differences across genders. Future hypoxic conditioning studies should further investigate these effect differences.”

- **It would also be nice to state that/if it was a single center study.**

Response: this was indeed a single-center study. We have now added this to the revised manuscript.

Manuscript changes (p. 8):

“Multiple randomized, double-blinded, and placebo-controlled N-of-1 trials were performed at Radboud University Medical Center, Nijmegen, the Netherlands. All participants underwent four distinct hypoxia and one placebo protocol in duplicate. This exposure to ten interventions per participant in total allowed us to compare treatment effects at the inter- and intra-individual level. This single-center study was approved by the Medical Research Ethics Committee East Netherlands, The Netherlands (reference number NL.77891.091.22) and has been registered at clinicaltrials.gov (ClinicalTrials.gov Identifier: NCT05214287).”

- **Regarding demographics, it was great to see a 1:1 female:male ratio.**

Response: we thank the reviewer for this comment. For completeness, we have now added an extra exploratory analysis for the differential effects that we observed between men:women subgroups, which demonstrates that women overall score higher in effect size compared to men.

Manuscript changes (Results, p. 5):

“Women reported significantly more symptomatic improvement on all three participant-rated symptoms compared to men (estimates between 0.39 and 0.90). Disease severity did not modify the observed effects.”

Discussion, p. 7:

“Interestingly, women in our cohort reported more positive results on all participant-rated outcomes compared to men. Women are generally underrepresented in PD trials and a better understanding of gender differences is imperative. Factors to possibly consider include the notion that striatal binding on DAT scans is generally higher in women compared to men throughout the disease course, and also that levodopa-induced dyskinesias are more common in women (acknowledging that this could result directly from relative overdosing in women)^{45,46}. Although speculative, it is possible that a gender-related difference in striatal sensitivity to the short-term effects of hypoxic conditioning might explain some of the observed effect differences across genders. Future hypoxic conditioning studies should further investigate these effect differences.”

- How was sleep apnea assessed before inclusion?

Response: sleep apnea was assessed by either confirmation through diagnosis by a somnologist, or a self-reported positive history during the screening phase (e.g., spouse reporting periods of breathing stops during sleep). This is now added to the manuscript.

Manuscript changes (p. 26):

"assessed by either confirmation through diagnosis by a somnologist or a self-reported positive history during the screening phase."

- Could the authors provide a flowchart of patient inclusion, screening failures as well as dropouts and the corresponding reasons?

Response: we added a comprehensive flowchart of the study to the main text.

Manuscript changes:

- Added Figure 2: Flowchart of study participants through the protocol (p. 21)

- Is smoking, vascular comorbidities, prior ischemic events taken into consideration?

Response: during screening, detailed history-taking was performed by a medical doctor, including recent smoking and cardiovascular abnormalities, most importantly arrhythmias and congestive heart failure. These were an exclusion criterion for this first hypoxic conditioning study in individuals with PD, and these are now also listed in the new *Supplementary Table 1*. Hypertension was not an exclusion criterion and this has now been added to the supplementary table as well.

Manuscript changes:

- "*Supplementary Table 1: Inclusion and exclusion criteria*" added
- *Table 1 extended with demographic characteristics*

- Would presence of dysautonomia be of interest to be taken into consideration in the stratification of patients, despite early phase design and safety primary outcome, since it is relevant to the rationale behind the intervention?

Response: we agree with the reviewer that this is of interest. As an exploratory extra analysis, we have stratified our cohort by H&Y stage for heart rate, heart rate variability and blood pressure responses. We have plotted the Figures to support this response (see Figures below). There were no significant differences for these physiological parameters based on disease severity. Because this is an extra analysis that was not pre-specified, we have decided to add merely a statement about these specific results to the manuscript, instead of further increasing the number of figures. We have added group-level figures for breathing frequency, heart rate, systolic blood pressure and diastolic blood pressure to the Supplementary Materials (Supplementary Figure 2, p. 29). In addition, we have now added a remark to the Discussion.

Manuscript changes (Results, p. 4)

"After stratification according to H&Y stage, there were no significant between-group differences in responses in systolic or diastolic blood pressure, heart rate or heart rate variability to mild or moderate levels of hypoxia (data not shown)."

Discussion:

"Furthermore, we could not establish difference in autonomic responses between disease severity groups, apart from a decreased hypoxic ventilatory response in people with PD. This absence of hypoxia-related AEs is in line with previous trials [...]"

- Added Supplementary Figure 2 (p. 29)

- Similarly, have the authors considered environmental exposures and genetic risk factors that are strongly related to hypoxia response pathway in PD?

Response: unfortunately, we do not have access to such data. Genetic testing is only performed in cases with suspected familial or young-onset forms of PD, and information on environmental exposures are only recently beginning to be collected in clinical care. We do agree that this would be interesting to include in future work, and have now included this in the protocol of planned follow-up studies in our center.

- Treatment with MAOB inhibitors has been suggested to modulate mitochondria homeostasis – is that taken into consideration in the selection of participants?

Response: there were no individuals with MAO-b inhibitors in the study population. Treatment with these compounds is relatively uncommon in the Netherlands. We have now added this to (the further expanded) *Table 1*.

Manuscript changes:

- Expanded *Table 1* to contain detailed demographic characteristics

- Trial participants have H&Y ranging from 1.5. to 3, thus reflecting a wide range of motor symptom severity. This may be an important parameter in the context of this type of intervention, so I was wondering if the authors have looked into the adverse event number and type in each H&Y category separately.

Response: this is an excellent point, which we have now specified in the manuscript. We have additionally added an explorative analysis to test whether the short-term symptomatic responses we identified differed based on disease severity. This was not the case.

Manuscript changes (Results, p. 4-5):

“AEs were most common in participants with H&Y 3 (average 5.7 AEs per participant, compared to 4.6 and 2.8 in H&Y 1.5-2 and H&Y 2.5, respectively). However, AE incidence rates were not significantly different between disease severity subgroups.”

[..]

“Women reported significantly more symptomatic improvement on all three participant-rated symptoms compared to men (estimates between 0.39 and 0.90). Disease severity did not modify the observed effects.”

- I understand that cognitive assessment could not be included at the day of each intervention, in OFF medication state, but it would be a very interesting parameter to know at baseline, as well as after the completion of the trial protocol (in ON medication state).

Response: We agree with the reviewer, but unfortunately, we have not collected these data.

- In comparison to studies in Alzheimer, the duration of intervention is shorter – compared to repeated intermittent hypoxia sessions daily, 3-5 days, for one or several weeks. Is there a specific rationale behind it?

Response: as this was the first hypoxic conditioning trial in Parkinson’s disease, we first set out to establish the safety and dose-response relation of several *individual* hypoxic protocols, from intermittent and continuous hypoxia, to mild to moderate stimuli. This is now clarified in the revised Introduction and Methods section.

Manuscript changes (Introduction, p. 3):

“In this phase 1 trial, we assessed the safety, feasibility, and short-term effects of different individual-session protocols of hypoxic conditioning in individuals with PD. This trial employed a double-blinded, randomized placebo-controlled multiple N-of-1 design to assess different hypoxic conditioning protocols in all participants using Bayesian and frequentist analyses. This design is especially useful as it supports the dose-finding character of this study, and allows for a randomized intervention order in every individual participant so that participants act as their own control, thus allowing for the comparison of (sub)acute symptom responses across interventions and within individuals.”

Methods, p. 9:

“The spectrum of included hypoxia protocols was informed by multiple dose-finding hypoxic conditioning studies and included both mild and moderate hypoxic triggers^{39,43,68-71}. Although the total hypoxic dosage in the intermittent hypoxia interventions is lower due to the in-between normoxic bouts, we were interested in comparing same-duration interventions of continuous hypoxia in this dose-finding study.”

- In secondary outcomes the authors state: “This post-intervention window (i.e. 30 minutes) overlaps with the first therapeutic window of conditioning effects”. In the corresponding reference (nr 69) this window is defined from minutes to 24 hours. One may suggest that it can have been too early to have measured treatment effect after only 30 minutes.

Response: we agree with the reviewer and have added these considerations to the revised manuscript.

Manuscript changes (p. 8):

“An important limitation of the study includes the short time frame of the effect measurement. For example, first window effects may occur up to 24 hours after hypoxic exposure⁶³, which are not covered by the present assessor-rated scales (such as MDS-UPDRS part III) and serum analyses, although the measurement of participant-reported symptom scores was extended beyond this window.”

- The exploratory outcome measures could have been expanded with markers of HIF-1 induction downstream effects, oxidative stress and mitochondrial function for a clearer view of biological response to intervention.

Response: although we included erythropoietin as a mechanistic marker downstream of HIF-1, this marker did not demonstrate significant differences between interventions. In the revision, we have added single molecule array (Simoa) analyses of NfL and GFAP, as well as enzyme-linked immunosorbent assay (ELISA) analysis of brain-derived neurotrophic factor (BDNF). Out of these, BDNF is also likely upregulated by hypoxia. Unfortunately, a thorough study of the effects on mitochondrial function and oxidative stress, preferably by studying effects on peripheral blood mononuclear cells (PBMCs), was not possible with our material. However, we have added this consideration to the Discussion as a future perspective and have added this to the follow-up study protocol.

Manuscript changes (p. 7-8, 11):

- Serum analyses of BDNF (using ELISA), GFAP, NfL (both using the novel single molecule array [Simoa] analysis technique) added.
- "As an additional acute mechanistic marker of neuroprotection, BDNF was included. Preclinical and clinical studies suggest acute effects of hypoxia on BDNF^{82,83} and BDNF levels are associated with symptom severity in PD, possibly reflecting reduced neuroplasticity.^{84,85} NfL and GFAP were included as acute markers of neuronal injury.⁸⁶ NfL and GFAP are both associated with symptom severity in PD.⁸⁷⁻⁹⁰, whereas exercise reduces NfL.⁹¹ Notably, both NfL and GFAP are induced by chronic intermittent hypoxia, such as occurring in sleep apnea.^{92,93}"

"On the other hand, activation of the hypoxia response pathway is linked to several PD risk genes, including LRRK2, DJ-1 and PINK1-Parkin, and might thereby have protective or compensatory properties.⁴⁷ Future studies could investigate the effects of hypoxic conditioning on mitochondrial function and oxidative stress, such as by studying peripheral blood mononuclear cells (PBMCs)."

Results

- Can the authors provide some more details on how the SAEs were assessed unlikely associated to intervention (except for the fall from stairs)? It is stated that the SAEs TIA and AF occurred after the placebo session, but was placebo the first session in the protocol, or had they performed any of the hypoxia sessions before that?

Response: we agree with the reviewer, and this is now further specified in the protocol.

Manuscript changes:

"These four were assessed as being unlikely to be related to the study intervention due to their timing and context. Both acute-onset severe AEs, i.e., TIA and atrial fibrillation, occurred after a placebo intervention. Atrial fibrillation occurred in a patient with (in retrospect) a positive history for palpitations, and occurred 1.5 weeks after the first hypoxia intervention. The TIA occurred three weeks after the screening procedure and one week after a placebo intervention."

**- Regarding efficacy, I suggest that the authors soften the statements regarding improvement, as placebo was similar or even slightly superior to hypoxia intervention in several outcome measures.
- Differences in self-reported symptoms do not seem to be consistent, or dose- or intervention type-related, neither are reflected to the rater assessments, which further precludes strong conclusions regarding efficacy.**

Response: we agree with the reviewer. We have checked the entire manuscript for such statements, and have carefully toned these down accordingly.

Manuscript changes (p. 5):

"Singles sessions of intermittent hypoxia at F_{iO_2} 0.163 improved participant-selected symptoms (0.57, 95% CI 0.23–0.92) and urge to take dopaminergic medication (0.48, 95% CI 0.11–0.86), but not global symptom impression (0.25, 95% CI -0.07–0.57). However, this improvement did not meet our predefined minimal clinically important difference (MCID) of 0.75."

"With regard to assessor-rated scales, hypoxia protocols did not show significant improvement in Movement Disorders Society Universal Parkinson's Disease Rating Scale (MDS-UPDRS) part III relative to placebo ($P > 0.05$). Although intermittent hypoxia at F_{iO_2} 0.163 reached the predefined MCID of 3.5 on the MDS-UPDRS part III, this difference was not significant ($P = 0.36$)."

Discussion (p. 6-7):

"Finally, the intermittent hypoxia protocol at F_{iO_2} 0.163 seemed most promising compared to the other hypoxia protocols for short-term improvement in participant-rated symptom scales, although the effects were modest. Also, assessor-rated assessment did not show significant acute improvement. With regard to mechanistic markers, no significant dose-dependent responses were identified."

Our secondary aim related to the effects of hypoxic conditioning on clinical outcomes, and we found that participant-rated symptom scales and MDS-UPDRS part III suggest modest short-term symptomatic effects of intermittent hypoxia at FIO₂ 0.163 relative to placebo and other conditioning protocols, although the effects were not consistent across outcome measures.

Discussion

- Would the authors consider elaborating on whether hypoxia induced pathology in PD is cause or consequence (or both) and how would that, in relation to this trial's results, inform the design of future trials?

Response: we thank the reviewer for this suggestion and have added our considerations for the role of hypoxia in PD pathophysiology to the Discussion.

Manuscript changes (p. 7):

"In vitro, PDGFR β reached peak expression after six hours of hypoxia, with no significant increase after 1-hour hypoxia^{58,59}. Cortisol only showed a marginal signal towards increase at the lowest F_IO₂ level, in accordance with earlier studies.⁶⁰⁻⁶² Therefore, short-term moderate hypoxia does not seem to affect hypothalamus-pituitary-adrenal axis activity in PD. This makes it unlikely that short-term symptomatic effects are mediated through stress systems such as the noradrenergic system, which is also implicated in PD symptom severity.⁶³ From a mechanistic point of view, some studies suggest that exercise and hypoxic conditioning have pathways in common, which we could not confirm for BDNF.^{14,64,65} Several studies demonstrate that longer-term deep (intermittent) hypoxia is detrimental for mitochondrial dysfunction⁶⁶ and even induces α -synuclein aggregation and neurodegeneration.²⁹ Respiratory dysfunction in PD might accelerate neuronal hypoxic injury especially in this subgroup.³¹ GFAP and NfL, as markers of neuronal stress and neural degeneration and potential biomarkers of PD progression^{67,68}, were not altered after single sessions of hypoxic conditioning, reflecting no acute neuronal injury within two hours after onset of hypoxia. It should be noted that the first studies with these novel biomarkers suggest that in traumatic brain injury, increase in serum becomes apparent only several hours post-injury^{69,70}. Future studies should examine whether these markers are affected after longer hypoxic conditioning protocols. On the other hand, activation of the hypoxia response pathway is linked to several PD risk genes, including LRRK2, DJ-1 and PINK1-Parkin.⁵¹ Therefore, activation of using specific hypoxic conditioning protocols might have protective or compensatory properties."

- The authors state: "Cortisol only showed a marginal signal towards increase at the lowest FIO₂ level, in accordance with earlier studies. Therefore, any short-term symptomatic effects are unlikely mediated through subacute stress responses". Is this sentence correct, since stress would be expected to affect PD symptoms negatively?

Response: we agree with the reviewer that this sentence is confusing. We have now modified it to better reflect our intended statement.

Manuscript changes:

"Therefore, short-term moderate hypoxia does not seem to affect hypothalamus-pituitary-adrenal axis activity in PD. This makes it unlikely that short-term symptomatic effects are mediated through stress systems such as the noradrenergic system, which is also implicated in PD symptom severity.⁵⁹"

Overall, it is a novel and important study with somewhat complicated protocol, among other reasons due to the number of interventions that cannot be truly "washed out", since hypoxia preconditioning may have long lasting biological effects. The protocol compliance is very good, and the results support the conclusion regarding safety.

Response: we thank the reviewer for their suggestions, which have certainly improved the manuscript.

Response to reviewers

Reviewer #1 (Remarks to the Author):

The authors have adequately addressed our concerns. There was a great deal of material and discussion in the reviewer responses that is helpful and the reshaped manuscript makes it much easier to pull things together.

Response:

We thank the reviewer for their help in improving this manuscript.

Reviewer #3 (Remarks to the Author):

In this revised manuscript the authors have carefully respond to all my previous concerns and suggestions and made the necessary changes in texts, figures, and tables. I have no further concern on this much improved and important work.

Response:

We thank the reviewer for their kind words and help in this manuscript.

Reviewer #4 (Remarks to the Author):

This manuscript reports a randomized, double-blinded, placebo-controlled phase 1 clinical trial assessing hypoxia intervention in patients with Hoehn-Yahr scale 1.5-3 indicating early to mod-stage Parkinson's Disease (PD). The subjects tolerated the hypoxia exposures, although several experienced adverse events, most of minimal concern. The rationale, methods and results are presented coherently, and the Discussion places the findings in context without overstating the implications. This study lays the groundwork for expanded clinical testing of hypoxia for PD.

The N-of-1 design is an important strength of this study. Both 45 min continuous and 5 x 5 min intermittent hypoxia were applied, at what could be considered mild (FIO₂ 0.163) and moderate (FIO₂ 0.127 or 0.133) intensities, in each subject, affording robust Bayesian and frequentist statistical analysis and comprehensive evaluation of hypoxia's safety and feasibility for PD therapy.

Regarding secondary outcomes, the hypoxia interventions did not produce clinically important improvements in PD symptoms, although the subjects, particularly the women, reported some relief in their symptoms. Although achieving symptomatic improvements was not the primary objective, substantial treatment effects might not be expected from only one session per week. The outcomes of this study will inform the design of the planned studies described in the last paragraph of the Discussion.

Response:

We thank the reviewer for their kind words and help in further improving this manuscript.

The modest increases in heart rate during hypoxia is physiological evidence that the hypoxia exposures impose very little stress. A comment along those lines could be added to the Discussion.

Response: *we have added this consideration to the Discussion.*

Manuscript changes (p. 6): *“Although continuous exposure to F_IO₂ 0.127 led to oxygen saturations below 80% for some participants, which was a stop criterion, this was not accompanied by any reported discomfort or other abnormal vital signs, supporting the notion that this intervention imposes limited physiological stress.”*

The more intense hypoxia exposures (F_IO₂ = 0.127) seemed to provoke adverse responses in some subjects. For their planned trials the authors might consider gradually intensifying the hypoxia exposures over the first several sessions from initial F_IO₂ of 0.163 to the ultimate F_IO₂ of 0.127. Doing so might provoke physiological adaptations improving the subject’s hypoxia tolerance, analogous to gradually increasing altitude to condition the body for mountain climbing.

Response: *we will consider this in the design of our upcoming studies and have added this to the Discussion. For our next study, we are also considering personalized hypoxia protocols.*

Manuscript changes (p. 7): *“Due to the individual differences in physiological responses to our hypoxia protocols, personalized dosing or gradually increasing levels of hypoxia may be considered.”*

Did any of the 95 adverse events occur during the sessions, as opposed to post-session? Were AEs monitored only during the 10-week study, or was monitoring extended a certain time post-study (beyond week 10?). That information would be valuable to physicians considering hypoxia treatment for their patients.

Response: *other than discomfort or dyskinesia due to prolonged sitting still, nearly all other adverse events were reported after the interventions. We have now noted this explicitly in the manuscript.*

Manuscript changes (p. 4): *“Nearly all adverse events were reported one hour up to three days after the intervention, apart from discomfort or dyskinesia due to prolonged immobilization OFF-medication during the intervention.”*

Another matter to consider is the possibility of synergy between hypoxia and PD medications, since they act by different and possibly complementary mechanisms.

Response: *we agree with the reviewer that this is an interesting way of thinking. In an earlier version of the manuscript we considered such a possibility, but due to insufficient evidence to support these mechanisms, we unfortunately had to decide to remove this from the manuscript as this was deemed to speculative.*

The Screening Procedure subsection of the Methods states that two subjects were retained in the study by raising the lower FIO₂ to 0.133, implying only those two subjects breathed the slightly higher FIO₂. However, supplemental Table 2 shows the lowest FIO₂ was 0.133 for eight subjects. Please report that information.

Response: *The reviewer notes correctly that there are more than these two individuals treated at FiO₂ 0.133. To further personalize the intervention, we modified the protocol in conjunction with the medical-ethical committee, which allowed all participants included after this protocol amendment to be included at either FiO₂ 0.127 or 0.133, depending on whether stop criteria (in this case, SaO₂ or pO₂) were met at FiO₂ 0.127 but not at FiO₂ 0.133 during the screening intervention.*

Manuscript changes: *“As a consequence, two individuals that were initially excluded, could now complete the protocol within stop criteria with this personalized dosage strategy, and six other individuals could now enter the protocol with interventions at F_IO₂0.133 instead of 0.127.”*

Minor comment: Please provide a legend for Figure 1.

Response: *this is now added to the Figure.*

Reviewer #5 (Remarks to the Author):

In this phase 1 randomized controlled trial, the authors employ a multiple N-of-1 trial design to evaluate the safety and feasibility of hypoxic conditioning in individuals with Parkinson's disease (PD). This methodological approach - both in terms of the individualized trial design and the associated statistical analyses - is relatively novel and complex within the PD field. However, this can be viewed as a strength of the study, as it introduces a potentially more sustainable and personalized framework for early-phase clinical trials in this population. The authors have made substantial revisions in response to reviewer feedback, resulting in a significantly improved manuscript. The authors' transparency in sharing portions of their statistical code and providing power analyses for future studies is also appreciated. The study's conclusions are well supported by the data, and the interpretation of the results is reasonable. The methodology is sound, and the additional information provided enhances the reproducibility of the work, as does the prior publication of the trial's rationale and protocol in an open-access format. Finally, the authors have also considered, investigated, and discussed sex differences, which adds to the comprehensiveness of the analyses.

Response:

We thank the reviewer for their time and efforts in reviewing this manuscript and helping to improve it with their suggestions.

I have only a couple of minor remaining comments

- LEDD scores can be added in Table 1 for completeness regarding symptomatic treatment.

Response: *this is now added to Table 1 (page 20).*

- In line 98-101, the authors state: "One participant dropped out after two interventions due to recurrence of paroxysmal atrial fibrillation (unlikely related to study procedures) and was replaced. This participant was not included in the secondary outcomes analysis (Figure 2)." However, Figure 2 shows 20 participants finally included, and 20 is the number stated throughout the report, so this dropout was not replaced, if I understand the statement correctly.

Response: *this dropout was replaced, so a total of 21 individuals started the protocol, but only 20 finished the protocol. Only these individuals were part of the secondary analyses. This is now clarified in the text.*

Manuscript changes (p. 3-4):

"One participant dropped out after two interventions due to recurrence of paroxysmal atrial fibrillation (unlikely related to study procedures) and was replaced. This participant was not included in the secondary outcomes analysis (Figure 2). Therefore, 20 individuals successfully completed all 200 interventions."

- In Figure 3, state with an asterisk or footnote which AEs are included under the term "other", as well as their frequency and intervention category.

Response: *this explanation is now added.*

Manuscript changes: *"'Other' adverse events included mostly self-limiting paresthesia in one or more extremity (6, occurring in both the placebo as active intervention group), vagal symptoms (2) and a sensitive throat (2)." (p. 22)*